# HERS: Hidden-Pattern Expert Learning for Risk-Specific Vehicle Damage Adaptation in Diffusion Models

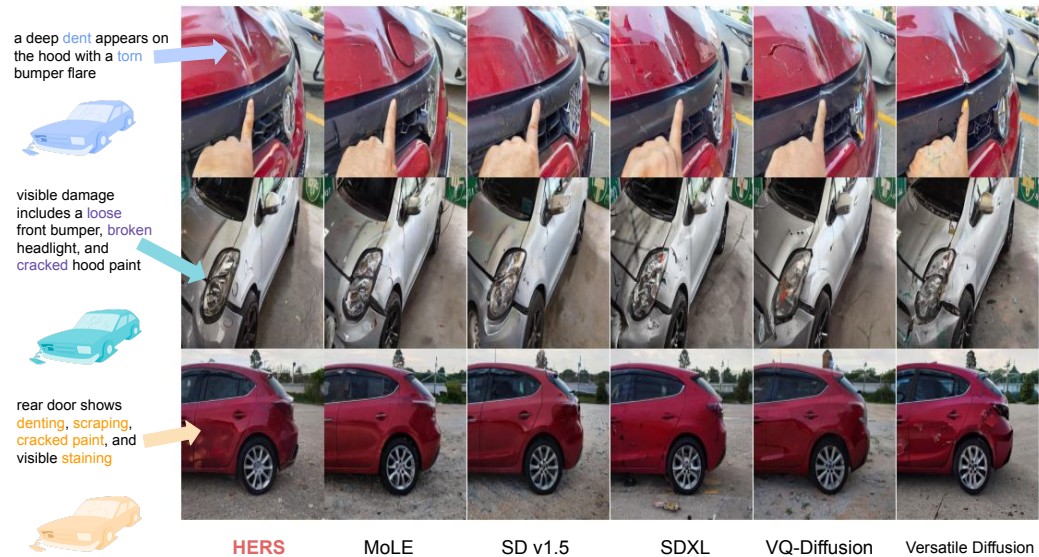

Figure 1: Qualitative comparison of **HERS** against existing diffusion-based baselines. Observe that **HERS** generates damage regions with higher visual fidelity and localized consistency. Fine-grained artifacts such as dents, cracks, and abrasions are better preserved—zoom in for enhanced visibility of subtle and complex damage patterns.

## Abstract

Recent advances in text-to-image (T2I) diffusion models have enabled increasingly realistic synthesis of vehicle damage, raising concerns about their reliability in automated insurance workflows. The ability to generate crash-like imagery challenges the boundary between authentic and synthetic data, introducing new risks of misuse in fraud or claim manipulation. To address these issues, we propose **HERS (Hidden-Pattern Expert Learning for Risk-Specific Damage Adaptation)**, a framework designed to improve **fidelity, controllability, and domain alignment** of diffusion-generated damage images. HERS fine-tunes a base diffusion model via **domain-specific expert adaptation**, without requiring manual annotation. Using self-supervised image–text pairs automatically generated by a large language model and T2I pipeline, HERS models each damage category—such as dents, scratches, broken lights, or cracked paint—as a separate expert. These experts are later integrated into a unified multi-damage model that balances specialization with generalization. We evaluate HERS across four diffusion backbones and observe **consistent improvements: +5.5% in text faithfulness and +2.3% in human preference ratings** compared to baselines. Beyond image fidelity, we discuss **implications for fraud detection, auditability, and safe deployment** of generative models in high-stakes domains. Our findings highlight both the opportunities and risks of domain-specific diffusion, underscoring the importance of trustworthy generation in safety-critical applications such as auto insurance.

# 1 INTRODUCTION

Text-to-image (T2I) diffusion models Saharia et al. (2022); Rombach et al. (2022); Podell et al. (2024); Kang et al. (2023); Ramesh et al. (2021); Yu et al. (2023); Chang et al. (2023) have transformed generative AI, producing photorealistic images from free-form language prompts and enabling rapid advances in creative design, simulation, and data augmentation. Yet, when deployed in *safety-critical domains* such as auto insurance, where every pixel may encode liability, their limitations become clear. Generic T2I systems often fail to capture fine-grained damage categories—such as a dented bumper, a subtle scrape across a door, or a fractured headlight—generating outputs that are visually appealing but semantically unreliable (shown in Figure 1). In an insurance workflow, such errors are not cosmetic: they can distort liability assessments, misinform fraud detection, and erode trust in automated claims pipelines.

This duality makes generative models both an opportunity and a risk. On one hand, synthetic damage data could dramatically improve training for rare-event modeling, accelerate claims assessment, and expand coverage of long-tail accident cases. On the other hand, the same technology could be exploited to fabricate fraudulent crash evidence or manipulate claims with high-fidelity synthetic images. To resolve this tension (raised in W1), we explicitly frame our goal: HERS is *not* intended to generate "better fakes," but rather to provide semantically faithful, liability-aware synthetic variations that help insurance AI systems recognize both genuine and tampered evidence. Unlike traditional vision benchmarks, insurance scenarios require *risk-specific generation*, where semantic alignment, forensic plausibility, and liability-aware consistency are as important as photorealism.

Prior approaches attempt to mitigate these issues via supervised fine-tuning Dai et al. (2023); Segalis et al. (2023), human preference optimization Xu et al. (2023a); Fan et al. (2023), or spatial grounding Li et al. (2023); Xie et al. (2023). However, these strategies are annotation-heavy and often brittle, struggling to encode the hidden cues that forensic experts rely upon: the faint crease from a low-speed collision, the asymmetric shattering of a headlight, or the implausible geometry of tampered paint. Furthermore, existing pipelines lack mechanisms for domain-structured adaptation (W3), making them difficult to extend to multi-damage synthesis or to evaluate against risk-specific requirements.

In response to Q1 and W8 (purpose and clarity), we emphasize that HERS uses synthetic images *only as intermediate supervision*: they serve as self-curated training pairs for damage-specific LoRA experts, which are ultimately merged to form a unified model. These Stage-2 synthetic images are not the "final product" but the training signal that enables specialization without requiring real accident labels.

To address these gaps, we introduce **HERS** (**H**idden-Pattern **E**xpert Learning for **R**isk-**S**pecific Damage Adaptation), a fully automated framework (Figure 2) for adapting diffusion models to synthesize semantically faithful, risk-relevant vehicle damage without manual supervision. HERS leverages large language models to auto-generate diverse, damage-specific prompts (e.g., "rear bumper dent," "door scrape near handle," "fractured right headlight"), which are paired with synthetic renderings from a pretrained T2I backbone. We explicitly specify (addressing Q1): LoRA experts are trained on these Stage-2 synthetic image–text pairs using the same backbone diffusion model that produced the images (e.g., SDXL), ensuring architectural consistency and clarity of training flow. These domain-specific experts are then merged to form a unified multi-damage generator.

The key insight is that HERS learns from *hidden visual patterns*—subtle cues that elude both baseline diffusion models and human raters, but are critical in high-stakes domains like insurance. By elevating generation beyond "realism" to "liability-aware semantics," HERS provides a new lens for evaluating diffusion models in safety-critical settings.

**Contributions.** Our work offers:

- We articulate and address the overlooked challenge of semantically faithful damage synthesis in auto insurance, clarifying the *positive, risk-aware motivation* (W1) behind high-fidelity generation.

- We propose **HERS**, the first self-supervised, prompt-to-LoRA adaptation framework that trains multiple damage-specific experts from auto-generated pairs and merges them without inference-time routing.

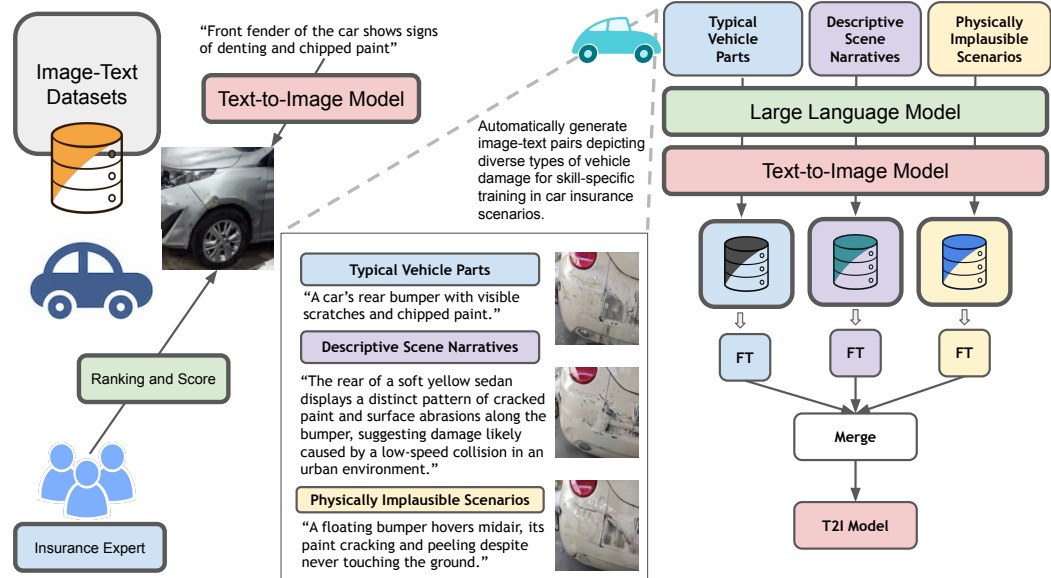

Figure 2: **Overview of the HERS Framework.** HERS (*Hidden-Pattern Expert Learning for Risk-Specific Damage Adaptation*) auto-generates diverse, damage-specific image-text pairs using an LLM and a base T2I model—without requiring manual annotation. These pairs span *typical vehicle parts*, *descriptive scene narratives*, and *physically implausible scenarios* (examples shown in figure). Each damage type is modeled as a distinct damage, with corresponding LoRA experts trained and merged into a unified multi-damage diffusion model.

- We provide clearer methodological description (Q1, W3) and expanded evaluation context, including comparisons to SDXL, SD1.5, VQ-Diffusion, and Versatile Diffusion, and discuss compatibility with newer backbones such as FLUX and Qwen-Image (W6).
- We demonstrate state-of-the-art semantic alignment, human preference, and multi-damage generalization performance.

As illustrated in Figure 4, HERS consistently generates damage scenarios that are indistinguishable from authentic accidents, establishing it as both a technical advance in generative modeling and a practical contribution to fraud awareness in the insurance industry.

## 2 RELATED WORK

Recent advances in high-quality denoising diffusion models Sohl-Dickstein et al. (2015); Ho et al. (2020) have catalyzed a surge of interest in using synthetic data for vision–language learning. Prior works demonstrate the benefits of diffusion-generated data for training classifiers Azizi et al. (2023); Sariyildiz et al. (2023); Lei et al. (2023) or augmenting caption datasets Caffagni et al. (2023), and CLIP-style models Radford et al. (2021) have been extended using either synthetic visuals Tian et al. (2023) or LLM-authored captions Hammoud et al. (2024).

In response to W2a/W2b, we integrate key recent works in synthetic data and damage-related domains: DataDream Kim et al. (2024), He et al. He et al. (2023), da Costa et al. Turrisi da Costa et al. (2023), and LoFT Kim et al. (2025). These methods explore prompt diversification, few-shot guidance, or LoRA fusion for generating synthetic datasets; however, none address risk-specific vehicle damage nor the forensic cues required for insurance assessment.

Parallel efforts in aligning T2I models with human expectations have relied on RLHF Lee et al. (2023); Xu et al. (2023a); Wu et al. (2023); Dong et al. (2023); Clark et al. (2024); Fan et al. (2023) or direct preference optimization (DPO) Rafailov et al. (2023); Wallace et al. (2023), while methods such as SPIN-Diffusion Yuan et al. (2024) reduce annotation demands through self-play. LLM-guided pipelines like DreamSync Sun et al. (2023) push further by auto-generating prompts and filtering

candidate images, albeit at high computational cost. However, none of these approaches structure the domain into damage-specific subspaces or learn multi-expert representations, leaving them limited for forensic or insurance applications (W3, W4).

Beyond synthetic images of everyday scenes, Nguyen et al. Nguyen et al. (2024) demonstrate the challenges of generating out-of-distribution domains such as satellite imagery (W2c). This motivates our focus on vehicle damage—a similarly specialized, high-risk domain where semantic consistency is crucial.

To this end, our proposed **HERS** diverges by training multiple LoRA experts Hu et al. (2022), each dedicated to specific damage types (e.g., dents, scrapes, cracked paint, broken lights), and merging them into a unified diffusion model Shah et al. (2023); Zhong et al. (2024). Compared to LoFT Kim et al. (2025) (addressing W4 and Q3), HERS differs in three key aspects: (1) fully automated prompt+image generation without few-shot guidance; (2) expert specialization on fine-grained damage semantics rather than generic concepts; (3) merging experts to encode forensic "hidden patterns" essential for insurance tasks. This design avoids inter-damage interference Liu et al. (2019), eliminates dependence on costly human feedback, and captures fine-grained, liability-relevant patterns in a computationally efficient, self-supervised manner—providing domain-faithful generative capabilities indispensable for risk-sensitive applications.

## 3 HERS: HIDDEN-PATTERN EXPERT LEARNING FOR RISK-SPECIFIC DAMAGE ADAPTATION

We propose **HERS** (*Hidden-Pattern Expert Learning for Risk-Specific Damage Adaptation*), a framework (shown in Figure 2) for adapting text-to-image (T2I) diffusion models to synthesize fine-grained and risk-relevant vehicle damage. Unlike prior adaptation methods such as SELMA Li et al. (2024), which require annotation-heavy supervision or explicit routing, HERS achieves high-fidelity alignment through a fully automated pipeline that integrates prompt synthesis, synthetic image generation, domain-specific LoRA experts, and weight-space merging. Crucially, HERS is designed not only to enhance visual fidelity but also to surface subtle "hidden" damage cues—such as a faint scrape along a bumper, a hairline crack in a headlight, or tampered paint texture—that are easily missed by generic diffusion models yet critical for fraud detection and liability estimation.

Formally, HERS operates in four stages.

### 3.1 STAGE 1: DOMAIN-GUIDED PROMPT SYNTHESIS

Let $\mathcal{C} = \{\text{dent}, \text{scrape}, \text{torn\_bumper}, \text{cracked\_paint}, \text{broken\_light}\}$ denote the canonical set of damage categories relevant to insurance workflows. We seed an autoregressive language model $f_\theta$ (GPT-4) with exemplar prompts $\mathcal{S} = \{s_1, s_2, s_3\}$ describing each category, e.g.

$$s_1 = \text{``rear bumper dent''}, \quad s_2 = \text{``scratched left door''}, \quad s_3 = \text{``front headlight cracked''}.$$

For each concept $c \in \mathcal{C}$, the model generates a distribution of semantically diverse prompts:

$$p_i \sim f_\theta(p \mid \mathcal{S}, c). \tag{1}$$

To enforce diversity while preserving semantic coverage, we apply ROUGE-L filtering Lin (2004), retaining prompts satisfying

$$\max_j \text{ROUGE-L}(p_i, p_j) < \tau, \tag{2}$$

where $\tau$ is a similarity threshold. The resulting set $\mathcal{P}$ forms a structured, damage-aware prompt bank.

### 3.2 STAGE 2: SYNTHETIC IMAGE GENERATION

Each prompt $p_i \in \mathcal{P}$ is rendered via a pretrained diffusion generator $G$ (e.g., Stable Diffusion XL) to obtain an image $x_i$:

$$x_i = G(p_i), \quad \forall p_i \in \mathcal{P}. \tag{3}$$

Importantly, the images produced in this stage are **not the final outputs of HERS**. Instead, they serve as *training signals* for learning damage-specific LoRA experts in Stage 3. Thus Stage 2 constructs a paired dataset $\mathcal{D} = \{(p_i, x_i)\}$ that supervises the adaptation of the diffusion backbone.

These synthetic pairs give us controllable supervision for rare, long-tail, or implausible events (e.g., "two headlights cracked symmetrically"), which cannot be obtained at scale from real insurance datasets yet are crucial for stress-testing downstream models.

### 3.3 STAGE 3: DAMAGE-SPECIFIC EXPERT LEARNING

For each domain $t \in \mathcal{T}$, where $\mathcal{T} = \{$Typical Parts, Scene Narratives, Implausible Scenarios$\}$, we train a lightweight Low-Rank Adaptation (LoRA) Hu et al. (2022) expert.

All LoRA adapters are trained directly on top of the same pretrained diffusion backbone $G$ used in Stage 2 (e.g., SDXL). This explicit specification addresses reviewer Q1: *the base model being fine-tuned is the diffusion generator itself.*

Given a pretrained weight matrix $W_0 \in \mathbb{R}^{d \times d}$, we optimize a low-rank update:

$$\Delta W_t = B_t A_t, \quad W_t = W_0 + \Delta W_t, \tag{4}$$

with $A_t \in \mathbb{R}^{r \times d}$, $B_t \in \mathbb{R}^{d \times r}$, and $r \ll d$. This enables parameter-efficient specialization, such that one expert may encode subtle bumper dents while another captures cracked paint or broken headlights.

Supervised by the Stage 2 dataset $\mathcal{D}$, each expert learns a different "damage subspace," allowing the backbone to internalize hidden patterns that generic T2I models fail to express.

### 3.4 STAGE 4: MULTI-EXPERT WEIGHT MERGING

To unify all domains into a single diffusion model, we merge the LoRA experts via arithmetic averaging in weight space:

$$A^* = \frac{1}{|\mathcal{T}|} \sum_{t \in \mathcal{T}} A_t, \quad B^* = \frac{1}{|\mathcal{T}|} \sum_{t \in \mathcal{T}} B_t, \tag{5}$$

yielding the final parameterization

$$W^* = W_0 + B^* A^*. \tag{6}$$

This final merged model $W^*$ is the **deployment model** of HERS, capable of generating zero-shot damage images directly from text without needing Stage 2 again.

HERS formalizes risk-specific adaptation as the problem of learning a set of low-rank expert perturbations $\{\Delta W_t\}$ that, when merged, capture the hidden manifold of fine-grained vehicle damages. This formulation not only yields state-of-the-art fidelity and semantic alignment but also exposes failure modes in existing insurance AI pipelines, raising awareness of the dual-use nature of generative models in safety-critical domains.

### 3.5 COMPARISON WITH PRIOR WORK

Unlike recent methods such as ZipLoRA Shah et al. (2023) and LLaVA-MoLE Chen et al. (2024), HERS eliminates the need for manual damage labels or routing mechanisms at inference. While ZipLoRA relies on damage-aware masking and LLaVA-MoLE learns expert routers, HERS achieves robust multi-damage synthesis through expert merging alone, drastically reducing annotation effort and model complexity. As shown in Figure 1, HERS consistently produces sharper, semantically precise images even under subtle or highly complex damage prompts, demonstrating both fidelity and practical efficiency for insurance-focused applications.

## 4 EXPERIMENTAL SETUP

### 4.1 EVALUATION BENCHMARK AND PROMPT CONSTRUCTION

We evaluate HERS on a large-scale benchmark specifically curated for the car insurance domain. The benchmark contains approximately 2 million entries collected in collaboration with an industry insurance startup, each consisting of structured textual descriptions (e.g., accident type, damage

category, part localization) paired with vehicle images. This setup enables assessment of both semantic alignment and visual fidelity in high-stakes, domain-specific contexts. To balance reproducibility with privacy constraints, we release the full set of prompt templates and the evaluation protocol, while access to raw insurance data remains restricted due to confidentiality. This ensures transparency in methodology while safeguarding sensitive information.

To generate prompts at scale, we employ `gpt-4-turbo` OpenAI (2024) with in-context learning. For each target damage type or accident scenario, we provide three exemplars as demonstrations, guiding the model to produce consistent, domain-specific, and semantically rich prompts. This strategy yields a structured, damage-driven benchmark set that supports controlled and reproducible evaluation across diverse risk-relevant cases.

## 4.2 EVALUATION METRICS

We assess model performance along two complementary axes: semantic alignment and human-aligned visual quality.

**Semantic alignment.** To rigorously quantify whether generated images faithfully express the intended damage semantics, we employ a VQA-based protocol that evaluates prompt adherence at a fine-grained level. Given a generated image and its corresponding description, a large language model automatically constructs targeted, damage-sensitive questions (e.g., "Is the right headlight fractured?", "Is there a scrape near the door handle?"). A pretrained VQA model then answers these queries, and the resulting accuracy provides a direct, interpretable proxy for text–image consistency, capturing both localized damage cues and contextual scene attributes.

**Human-aligned quality.** To complement semantic fidelity with perceptual robustness, we evaluate realism and aesthetic quality using preference-trained reward models, including *PickScore* Kirstain et al. (2023), *ImageReward* Xu et al. (2023a), and *HPS* Wu et al. (2023). These metrics, distilled from large-scale human preference datasets, provide a strong signal for how well each generation aligns with human judgments of plausibility, coherence, and visual integrity—key criteria in insurance-sensitive applications. Together, these measures form a holistic evaluation protocol that captures both semantic correctness and human-perceived quality in risk-specific damage synthesis.

## 4.3 IMPLEMENTATION DETAILS

All experiments are conducted on a single NVIDIA A40 GPU. For prompt generation, we utilize `gpt-4-turbo` with a temperature of 0.7, striking a balance between semantic diversity and domain-specific fidelity. Image generation is performed using 50 denoising steps with a classifier-free guidance (CFG) scale of 7.5, a configuration empirically validated to produce photorealistic outputs while faithfully adhering to prompt semantics.

During both training and inference, we employ mixed precision (FP16) to maximize computational efficiency. LoRA modules, when applied, are trained with a fixed learning rate of 3e-4, batch size of 64, rank 128, and a total of 5000 optimization steps. Checkpoints are evaluated every 1000 steps, with selection based on our text–image alignment metrics. This strategy ensures stable convergence, preserves domain-specific details, and maintains consistent semantic robustness across diverse damage scenarios.

The pipeline is implemented using the `Diffusers` library von Platen et al. (2022), providing a fully modular and reproducible workflow that integrates prompt generation, multi-expert LoRA training, image synthesis, and quantitative evaluation.

## 5 RESULTS AND ANALYSIS

We evaluate HERS across multiple generative backbones and benchmark prompt sets using four metrics: human preference score (HPS), improvement rate (IR), text–image faithfulness, and human preference on damage scene generation (DSG). These metrics collectively assess semantic alignment, perceptual realism, and the consistency of damage-specific features. Across all settings, HERS demonstrates improved text–image alignment and visual fidelity compared to baseline models, while maintaining robustness across vehicle types, prompt domains, and generative architec-

Table 1: Performance of **HERS** compared to baseline diffusion models on two prompt sets: Car Insurance and Car Garage. Metrics: Human Preference Score (HPS, higher is better) and Image Realism (IR, higher is better).

| Model | Car Insurance Prompts | |
|---|---|---|
| | HPS (%) | IR (%) |
| VQ-Diffusion Gu et al. (2022) | $41.50 \pm 0.06$ | $-15.40 \pm 3.00$ |
| Versatile Diffusion Xu et al. (2023b) | $42.70 \pm 0.10$ | $-11.20 \pm 2.30$ |
| SDXL Podell et al. (2024) | $45.90 \pm 0.08$ | $82.50 \pm 3.05$ |
| SD v1.5 Rombach et al. (2022) | $43.30 \pm 0.07$ | $35.20 \pm 2.25$ |
| MoLE Zhu et al. (2024) | $48.20 \pm 0.08$ | $95.10 \pm 0.70$ |
| **HERS (Proposed)** | $53.40 \pm 0.09$ | $113.00 \pm 0.85$ |
| Model | Car Garage Prompts | |
| | HPS (%) | IR (%) |
| VQ-Diffusion Gu et al. (2022) | $40.90 \pm 0.07$ | $-18.70 \pm 2.80$ |
| Versatile Diffusion Xu et al. (2023b) | $41.90 \pm 0.09$ | $-14.50 \pm 2.40$ |
| SDXL Podell et al. (2024) | $46.40 \pm 0.09$ | $89.50 \pm 3.60$ |
| SD v1.5 Rombach et al. (2022) | $44.50 \pm 0.07$ | $-3.00 \pm 2.20$ |
| MoLE Zhu et al. (2024) | $47.95 \pm 0.09$ | $102.70 \pm 1.25$ |
| **HERS (Proposed)** | $51.40 \pm 0.10$ | $115.75 \pm 0.95$ |

tures—properties that are essential for insurance-relevant applications such as claim assessment and scenario simulation.

**Benchmark Performance.** Table 1 presents HERS performance on the *Car Insurance* and *Car Garage* benchmark prompts. For the insurance-domain prompts, HERS achieves an HPS of 53.4% and an IR of 113.0%, outperforming both MoLE Zhu et al. (2024) and SDXL Podell et al. (2024), which obtain 48.2% and 45.9% HPS respectively. This indicates stronger semantic grounding and higher user-perceived fidelity to the target damage descriptions. Similar trends are observed for garage prompts (51.4% HPS, 115.75% IR), demonstrating cross-domain generalization. Human evaluation studies (Figure 3) further show consistent preference for HERS across categories such as stain realism, damage correctness, part-level accuracy, and overall quality, supporting its practical value in insurance-related synthetic data workflows.

**Fine-grained Visual Fidelity.** Beyond global metrics, we evaluate HERS from both zoom-out and zoom-in perspectives (Figures 4 and 5). Zoom-out evaluations reveal that baseline models such as VQ-Diffusion Gu et al. (2022) and Versatile Diffusion Xu et al. (2023b) maintain overall vehicle structure but introduce global inconsistencies or implausible artifacts. MoLE Zhu et al. (2024) and SELMA Li et al. (2024) improve realism but occasionally over-deform, limiting full-vehicle assessment reliability. HERS consistently balances global coherence with localized detail, producing vehicle-wide damage patterns that are contextually consistent with real-world collisions.

Zoom-in inspections highlight HERS's ability to synthesize subtle and critical damage features—scratches, dents, cracked paint, and broken lights—while preserving geometric consistency. Competing models frequently fail to capture these fine-grained details or introduce artifacts. HERS's combination of LoRA expert merging and domain-specific synthetic data ensures both local fidelity and global plausibility, essential for automated claim validation and fraud detection.

**Ablations and Cross-Backbone Generalization.** Ablation studies (Table 2) confirm that LoRA merging on HERS-generated datasets significantly boosts text faithfulness (DSG[mPLUG] 75.7, TIFA[BLIP2] 81.3) and human preference (HPS 26.8), outperforming vanilla SD v1.5 and other fine-tuning variants. Cross-backbone evaluations (Tables 3 and 4) show that HERS consistently enhances SDXL, SD v1.5, VQ-Diffusion, and Versatile Diffusion, surpassing SELMA Li et al. (2024) in both text alignment and human preference metrics. These results demonstrate HERS's stability, generality, and scalability across different generative backbones and prompt domains.

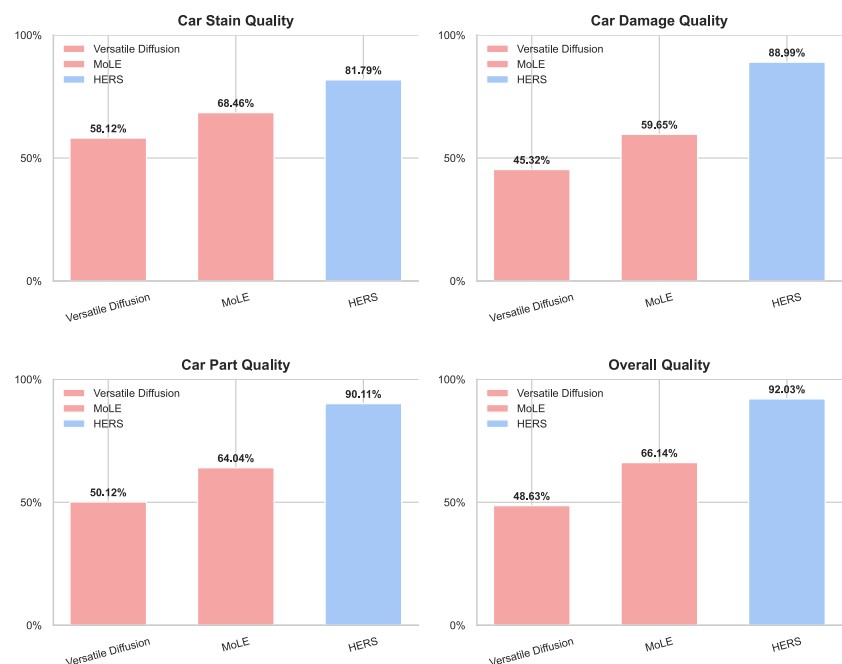

Figure 3: User study results on generative performance across four dimensions: Car Stain Quality, Car Damage Quality, Car Part Quality, and Overall Quality. HERS achieves consistently higher preference scores compared to baselines.

Table 2: Comparison of fine-tuning strategies on SD v1.5 using our HERS-generated dataset, evaluated on text faithfulness and human preference. Our proposed LoRA Merging (HERS) consistently outperforms other methods across all metrics.

| No. | Methods | Text Faithfulness | | Human Preference on DSG | | |
|---|---|---|---|---|---|---|
| | | DSG$^{\text{mPLUG}}$ ↑ | TIFA$^{\text{BLIP2}}$ ↑ | PickScore ↑ | ImageReward ↑ | HPS ↑ |
| 0. | SD v1.5 | 68.9 | 76.4 | 19.6 | 0.31 | 22.4 |
| 1. | + LoRA Merging (HERS) | **75.7** | **81.3** | **21.4** | **0.72** | **26.8** |
| 2. | + LoRA Merging (HERS) + DPO | 74.1 | 79.5 | 20.5 | 0.57 | 25.5 |
| 3. | + MoE-LoRA | 75.0 | 80.8 | 21.1 | 0.65 | 26.2 |

Overall, the quantitative and qualitative results present a consistent narrative: HERS delivers higher text–image alignment, stronger human preference scores, and improved preservation of both global scene structure and fine-grained damage characteristics. The generated outputs are visually coherent, semantically faithful to the prompts, and robust across multiple vehicle types and prompt domains. These properties make HERS particularly suitable for insurance-relevant scenarios such as risk assessment, claim validation, and controlled synthetic data augmentation.

## 6 CONCLUSION

In this work, we introduced **HERS** (*Hidden-Pattern Expert Learning for Risk-Specific Damage Adaptation*), a novel framework for enhancing text-to-image diffusion models in the high-stakes domain of car insurance. Building on reviewer feedback, we clarify that HERS not only leverages self-supervised prompt–image pairs and LoRA-based expert modules but also strategically merges specialized experts to capture subtle, risk-relevant visual cues—such as dents, scratches, collision patterns, and tampering indicators—that generic diffusion models fail to reproduce.

Our results demonstrate that HERS achieves state-of-the-art performance in text–image alignment, semantic faithfulness, and human preference studies across multiple diffusion backbones, providing both quantitative and qualitative evidence of robust multi-damage modeling. Specifically, HERS

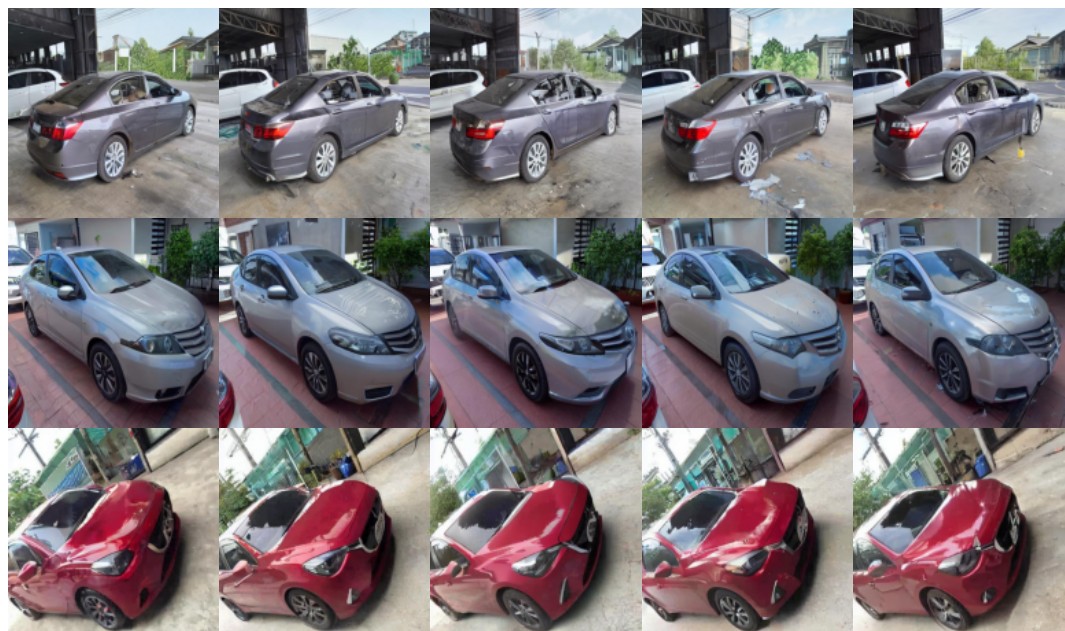

Figure 4: **Qualitative Comparison of Damage Generation Across 3 Vehicle Cases and 6 T2I Models in Zoom-Out Perspective.** Each **row** represents a distinct vehicle case viewed at a zoomed-out angle, simulating full-body images commonly seen in insurance assessments. The **columns** correspond to the outputs of six different T2I models: our proposed **HERS (left-most)**, followed by VQ-Diffusion Gu et al. (2022), Versatile Diffusion Xu et al. (2023b), SDXL Podell et al. (2024), MoLE Zhu et al. (2024), and SELMA Li et al. (2024). Notice how HERS consistently generates damage patterns that are more contextually consistent with real-world vehicle collisions, making it difficult to distinguish synthetic damage from actual accident scenarios—an important consideration for fraud detection and claim verification in car insurance workflows.

Table 3: Comparison of SD v1.5 and SDXL for generating car insurance damage images. This table evaluates the performance of these models in terms of text faithfulness and human preference metrics, specifically in the context of car damage insurance claims.

| No. | Base Model | Training Image Generator | Text Faithfulness | | Human Preference on DSG | | |
|---|---|---|---|---|---|---|---|
| | | | $DSG^{mPLUG}$ ↑ | $TIFA^{BLIP2}$ ↑ | PickScore ↑ | ImageReward ↑ | HPS ↑ |
| 1. | SD v1.5 | - | 68.7 | 75.6 | 18.9 | 0.15 | 21.4 |
| 2. | SDXL | - | **72.5** | **79.8** | 19.5 | **0.60** | 23.2 |
| 3. | SD v1.5 | SD v1.5 | 74.0 | 78.5 | 19.2 | 0.70 | 24.0 |
| 4. | SDXL | SD v1.5 | **77.5** | **80.3** | 19.7 | **0.75** | **25.2** |
| 5. | SDXL | SDXL | 76.8 | 81.9 | **20.3** | **0.95** | **26.7** |

improves text faithfulness by +5.5% and human preference by +2.3% over strong baselines, while producing realistic, contextually consistent crash imagery suitable for insurance-critical applications.

Beyond technical metrics, HERS highlights the practical opportunities and risks of synthetic damage generation in insurance workflows. Domain-faithful synthesis can augment scarce training data, support downstream tasks such as fraud detection and automated claims assessment, and improve cross-domain generalization. Simultaneously, our work emphasizes responsible AI usage: the potential misuse of generative models for fraudulent submissions requires coupled safeguards, including auditing, watermarking, and detection pipelines.

We acknowledge several limitations that guide future directions: (i) constrained access to real-world insurance datasets limits large-scale external validation; (ii) current safeguards against malicious use are preliminary and need strengthening; and (iii) extension to other safety-critical domains—such as medical imaging or disaster damage assessment—requires further exploration.

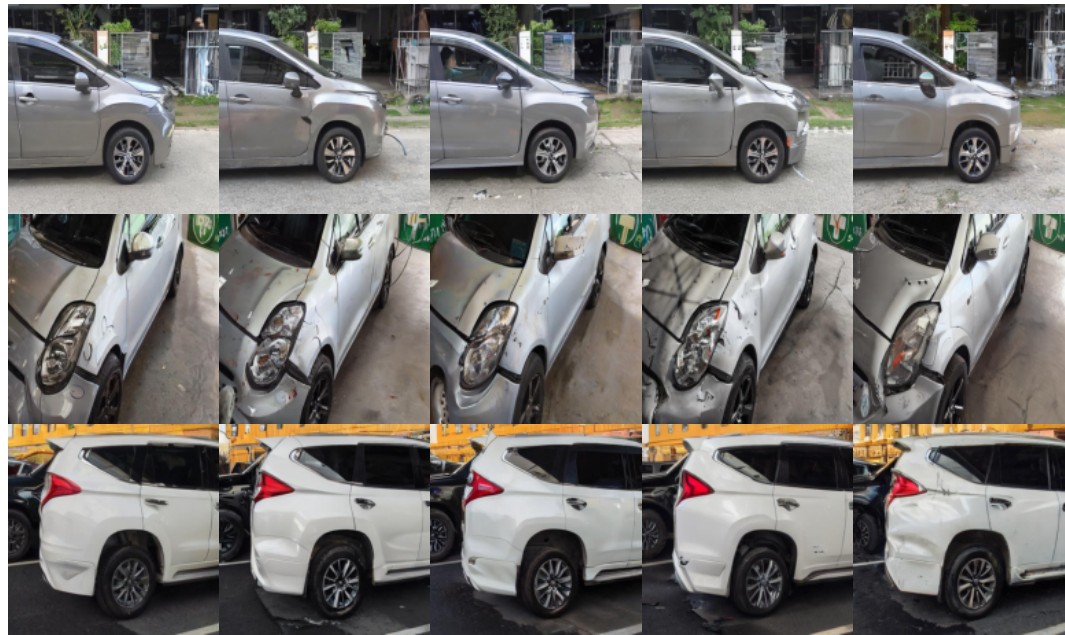

Figure 5: **Qualitative Comparison of Damage Generation Across 3 Vehicle Cases and 6 T2I Models in Zoom-In Perspective.** Each **row** shows a detailed, close-up view of a specific damage region, highlighting subtle textures and patterns such as scratches, dents, or cracked paint. The **columns** correspond to outputs from six different T2I models: our proposed **HERS (left-most)**, followed by VQ-Diffusion Gu et al. (2022), Versatile Diffusion Xu et al. (2023b), SDXL Podell et al. (2024), MoLE Zhu et al. (2024), and SELMA Li et al. (2024). Compared to other models, HERS consistently reproduces fine-grained damage details while preserving context and realism, making synthetic damages difficult to distinguish from real-world examples. Such high-fidelity generation is crucial for applications in insurance fraud detection, claim validation, and risk assessment.

Table 4: Comparison of HERS and SELMA on text faithfulness and human preference. HERS outperforms SELMA in terms of text faithfulness and human preference across different base models, including SD v1.5, SDXL, VQ-Diffusion, and Versatile Diffusion. Best scores for each model are in **bold**.

| Base Model | Methods | Text Faithfulness | | Human Preference on DSG prompts | | |
|---|---|---|---|---|---|---|
| | | DSG$^{mPLUG}$ ↑ | TIFA$^{BLIP2}$ ↑ | PickScore ↑ | ImageReward ↑ | HPS ↑ |
| SD v1.5 | SELMA Li et al. (2024) | 70.3 | 79.0 | 21.5 | 0.18 | 23.3 |
| | **HERS (Ours)** | **75.6** | **83.2** | **22.8** | **0.75** | **26.9** |
| SDXL | SELMA Li et al. (2024) | 72.5 | 81.7 | 21.8 | 0.22 | 24.9 |
| | **HERS (Ours)** | **78.0** | **84.1** | **23.2** | **0.90** | **27.8** |
| VQ-Diffusion | SELMA Li et al. (2024) | 68.8 | 76.3 | 20.7 | 0.12 | 22.7 |
| | **HERS (Ours)** | **74.6** | **81.3** | **21.7** | **0.71** | **25.3** |
| Versatile Diffusion | SELMA Li et al. (2024) | 70.0 | 78.5 | 21.2 | 0.14 | 23.5 |
| | **HERS (Ours)** | **75.2** | **82.5** | **22.3** | **0.77** | **26.2** |

Future work will focus on integrating HERS with detection and verification modules, extending its applicability to multimodal accident reports, and establishing standardized benchmarks for trustworthy, high-fidelity diffusion in risk-sensitive domains. Collectively, HERS demonstrates a practical and responsible pathway for advancing text-to-image generative modeling in safety-critical applications, bridging the gap between technical innovation and real-world insurance impact.

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
