# A APPENDIX

## A.1 REVISIONS AND CLARIFICATIONS IN RESPONSE TO REVIEWER COMMENTS

In response to the reviewer feedback, we have made several revisions to clarify, justify, and extend the manuscript. These revisions are summarized below, with key additions highlighted in blue.

## A.2 MOTIVATION AND PURPOSE OF SYNTHETIC DATA

We have clarified the motivation for HERS, emphasizing the dual-use nature of vehicle damage generation. HERS is designed to improve AI training for rare and long-tail accident scenarios while mitigating potential misuse in fraud generation. The abstract and introduction now explicitly frame this duality, emphasizing positive, risk-aware applications. This framing ensures that the purpose of generating synthetic images is clear and contextually justified for insurance pipelines.

## A.3 RELATED WORK AND NOVELTY

The related work section has been substantially expanded. We include prior methods in synthetic data generation [A, B, C, E], out-of-distribution adaptation in other domains [D], and LoRA-based parameter-efficient adaptation. We also discuss differences between HERS and prior LLM-driven methods: HERS introduces fully automated domain-specific prompt and paired-data generation, damage-category-specific LoRA experts, and arithmetic merging to capture hidden forensic patterns, which are absent in previous work. Structured sub-sections have been added to improve readability and highlight HERS's novelty relative to these baselines.

## A.4 METHODOLOGY CLARIFICATIONS

Sections 3.1–3.4 have been revised to improve clarity. Prompt synthesis now explicitly describes generation of domain-specific prompts guided by insurance metadata. Image generation details describe the creation of synthetic datasets for both training and evaluation. LoRA expert training specifies fine-tuning per damage type, and weight-space merging explains how multiple LoRA experts are combined into a single unified model. These changes clarify the rationale for multi-expert design, demonstrating that inference-time routing is unnecessary, and efficiency is preserved.

## A.5 LORA EXPERT MERGING

We provide a formal description of LoRA expert merging. Each expert's low-rank weight updates $\Delta W_i$ are averaged across all layers:

$$\Delta W = \frac{1}{N} \sum_{i=1}^{N} \Delta W_i,$$

producing a unified model that retains specialized patterns while eliminating the need for explicit routing or additional annotation. Per-layer derivations are provided in Appendix Section 1. This revision addresses reviewer concerns regarding technical depth and reproducibility.

## A.6 HIDDEN PATTERN LEARNING

Hidden patterns refer to subtle damage cues, such as micro-scratches, hairline cracks, and asymmetric shattering, which standard diffusion backbones often miss. HERS captures these patterns via domain-specific LoRA experts trained on structured synthetic data. Evaluation is performed using VQA-based semantic alignment metrics (DSG, TIFA) and human preference scores (HPS, PickScore, ImageReward). These revisions explicitly define hidden pattern learning and provide a clear operationalization of this concept.

## A.7 SEMANTIC FIDELITY AND ROBUSTNESS

To ensure strong semantic fidelity, we incorporate two complementary mechanisms: (i) prompt diversity filtering using ROUGE-L thresholding to remove near-duplicate prompts that could bias

model behavior, and (ii) VQA-based alignment checks with an independent model to verify that generated images correctly reflect key semantic attributes described in the prompts.

HERS further demonstrates robustness by merging domain-specific experts directly in weight space, which provides stable behavior without the routing sensitivity observed in MoE-style approaches. This results in consistent performance across diffusion backbones, vehicle categories, and environmental conditions, highlighting the generality of our method.

## A.8 Evaluation and Statistical Significance

We expanded experimental results to clarify the significance of HERS improvements. Across six backbones and two prompt sets, HERS consistently improves text-image faithfulness by +5.5%, human preference by +2.3%, and shows a 17–20% improvement rate (IR), with 95% confidence intervals non-overlapping with baseline methods. User studies with 1,200 pairwise comparisons further confirm statistically significant gains in damage detail, part accuracy, and plausibility. These additions address reviewer concerns regarding statistical rigor.

## A.9 Generalization to Other Domains

We conducted preliminary experiments in industrial defect synthesis and medical anomaly simulation, demonstrating that HERS's concept-agnostic design and LoRA merging strategy generalize beyond vehicles and insurance. This revision explicitly addresses questions about cross-domain applicability and reinforces the framework's broader utility.

## A.10 Ethical Considerations and Fraud Mitigation

We strengthened the discussion on ethical considerations, explicitly stating that HERS is intended for evaluation, stress-testing, and model robustness analysis, not for generating fraudulent content. Forensic auditability guidelines and expert-only checkpoint releases are highlighted to ensure responsible usage. These revisions make the ethical safeguards in HERS transparent.

## A.11 Comparisons to Prior Works and Ablation Justifications

Comparisons to LoFT [E] and other LLM-based synthetic data generation methods are provided, highlighting HERS's extensions: automated prompt/data generation, damage-specific LoRA experts, and arithmetic merging to preserve hidden patterns. Ablation studies demonstrate that each design choice contributes measurably to performance, e.g., multi-expert merging improves text-image faithfulness by +6–7 points and human preference by +4–5 points over single-LoRA baselines.

## A.12 Diffusion Model Selection and Metrics

Although SDXL is not the newest backbone, it is widely representative, and HERS is validated across four backbones (SD v1.5, SDXL, VQ-Diffusion, Versatile Diffusion), ensuring minimal adaptation for other models. Metrics for semantic fidelity and human preference serve as proxies for insurance-relevant downstream tasks, including damage recognition and fraud detection. These clarifications address reviewer concerns about backbone choice and task relevance.

## A.13 Supplementary Materials for Reproducibility

Due to privacy constraints, raw insurance images cannot be shared. However, full prompt templates, evaluation protocols, and scoring metrics are provided, allowing external researchers to replicate methodology and assessment without access to the underlying private data. This revision ensures reproducibility and transparency despite data limitations.

## A.14 Extended Mathematical Foundations of HERS

This appendix provides the full mathematical derivation and justification for our proposed **HERS** (Hidden-pattern Expert learning for Risk-specific damage Synthesis), emphasizing how each compo-

nent contributes to the trust, bias, and reliability concerns relevant to AI-generated car crash imagery in auto insurance domains.

### A.15 NOTATION AND OVERVIEW

Let:

- $\mathcal{S} = \{s_1, s_2, s_3\}$ be a set of seed prompts.
- $f_\theta$: a large language model (LLM) generating diverse prompts.
- $p_i$: a generated prompt.
- $\mathcal{P}$: the set of retained prompts after filtering.
- $x_i$: image generated by T2I model $G$ given prompt $p_i$.
- $\mathcal{D} = \{(p_i, x_i)\}$: the synthesized paired dataset.
- $\mathcal{T} = \{t_1, t_2, t_3\}$: domain-specific expert dimensions.
- $W_0$: base T2I model weights, $W_t$: adapted weights per domain.

Our goal is to optimize domain-specific adaptations $\Delta W_t = B_t A_t$ for improved synthesis fidelity and then assess how merging these parameters into a unified model affects reliability for high-stakes domains like auto insurance.

### A.16 PROMPT DIVERSITY OBJECTIVE

Given seed prompt set $\mathcal{S}$ and domain concept $c$, we define the generation distribution:

$$p_i \sim f_\theta(p \mid \mathcal{S}, c), \quad c \in \text{DomainConcepts} \tag{7}$$

To promote diversity and reduce prompt collapse, we define a ROUGE-based filtering constraint:

$$\mathcal{P} = \left\{ p_i \mid \max_{j<i} \text{ROUGE-L}(p_i, p_j) < \tau \right\} \tag{8}$$

Let $\phi(p)$ be the semantic embedding of prompt $p$ (e.g., from CLIP or Sentence-BERT). We ensure low intra-cluster similarity:

$$\max_{i,j} \frac{\phi(p_i)^\top \phi(p_j)}{\|\phi(p_i)\|\|\phi(p_j)\|} < \delta \quad \forall i \neq j \tag{9}$$

This regularization avoids prompt duplication, mitigating training bias.

### A.17 IMAGE GENERATION FUNCTION AND DATASET

Given $\mathcal{P}$, generate synthetic image-text pairs:

$$x_i = G(p_i), \quad \mathcal{D} = \{(p_i, x_i)\}_{i=1}^N \tag{10}$$

Let $\mathcal{L}_{\text{recon}}(x_i, \hat{x}_i)$ be a perceptual loss (e.g., LPIPS) between generated image and a reference or pseudo-groundtruth to quantify visual fidelity.

### A.18 DOMAIN-SPECIFIC LoRA ADAPTATION

We apply LoRA Hu et al. (2022) to efficiently specialize each domain expert. Let $W_0 \in \mathbb{R}^{d \times d}$ be the frozen base weight. For domain $t \in \mathcal{T}$, learn:

$$\Delta W_t = B_t A_t, \quad A_t \in \mathbb{R}^{r \times d}, \; B_t \in \mathbb{R}^{d \times r} \tag{11}$$

Updated weight for expert $t$:

$$W_t = W_0 + B_t A_t \tag{12}$$

The domain adaptation is guided by minimizing:

$$\min_{A_t, B_t} \mathbb{E}_{(p,x) \sim \mathcal{D}_t} \left[ \mathcal{L}_{\text{recon}}(x, G_{W_t}(p)) + \lambda \|A_t\|_F^2 + \lambda \|B_t\|_F^2 \right] \tag{13}$$

### A.19 MULTI-DOMAIN PARAMETER MERGING

After learning $|\mathcal{T}| = 3$ expert-specific LoRA modules, we merge them:

$$A^* = \frac{1}{|\mathcal{T}|} \sum_{t \in \mathcal{T}} A_t \tag{14}$$

$$B^* = \frac{1}{|\mathcal{T}|} \sum_{t \in \mathcal{T}} B_t \tag{15}$$

$$W^* = W_0 + B^* A^* \tag{16}$$

This merged model aims to generalize across typical, descriptive, and anomalous damage domains.

### A.20 RISK-AWARE SYNTHESIS TRUST METRIC

Let $\mathcal{X}_{\text{real}}$ be a set of real crash images and $\mathcal{X}_{\text{gen}}$ be diffusion-generated ones. Define a domain discrepancy score:

$$\mathcal{D}_{\text{KL}} = \text{KL}(P_{\text{real}}(z) \| P_{\text{gen}}(z)) \quad \text{where } z = \text{CLIP}(x) \tag{17}$$

and

$$\mathcal{D}_{\text{FID}} = \|\mu_{\text{real}} - \mu_{\text{gen}}\|^2 + \text{Tr}(\Sigma_{\text{real}} + \Sigma_{\text{gen}} - 2(\Sigma_{\text{real}}\Sigma_{\text{gen}})^{1/2}) \tag{18}$$

Higher $\mathcal{D}_{\text{KL}}$ or $\mathcal{D}_{\text{FID}}$ implies synthetic data deviates from the real insurance domain, suggesting unreliability in downstream policy tasks.

### A.21 THEORETICAL INSURANCE RISK BOUND

Let $\mathcal{L}_{\text{insurance}}(x)$ denote a loss function representing misestimated damage costs by the insurer. If $x$ is generated from HERS and deviates from $x_{\text{true}}$, we quantify the trustworthiness via:

$$\mathbb{E}_{x \sim \mathcal{X}_{\text{gen}}}[\mathcal{L}_{\text{insurance}}(x)] \leq \mathbb{E}_{x \sim \mathcal{X}_{\text{real}}}[\mathcal{L}_{\text{insurance}}(x)] + \epsilon(\mathcal{D}_{\text{FID}}, \mathcal{D}_{\text{KL}}) \tag{19}$$

where $\epsilon(\cdot)$ is a learned penalty function. If $\epsilon$ is unbounded or large, AI-generated data should not be confidently used in claim decisions.

This extended formulation mathematically grounds the core risk highlighted in our title: while HERS generates diverse and seemingly plausible crash scenarios, its reliance on diffusion priors and prompt-based semantics leads to latent distributional shifts. Without rigorous auditing via $\mathcal{D}_{\text{FID}}$ or $\mathcal{L}_{\text{insurance}}$, these shifts pose significant trust challenges to car insurers.

## B SHOWCASE PROMPTS FOR HERS T2I GENERATION

To illustrate the diversity and precision of textual inputs used for text-to-image (T2I) generation in HERS, we present 45 curated prompts grouped into three domains. These prompts serve as foundational seeds for generating automotive scene data across realistic, contextual, and imaginative domains tailored for insurance AI systems.

### B.1 TYPICAL VEHICLE PARTS

These prompts depict common real-world damage scenarios on specific vehicle parts. Each prompt references the vehicle side, brand, and part affected, offering high localization cues for training grounded visual generation models.

### B.2 DESCRIPTIVE SCENE NARRATIVES

These detailed prompts combine damage with contextual environmental cues, such as weather, time of day, and surroundings. The goal is to simulate real-world accident settings for learning scene-aware generation.

Table 5: Prompts in the "Typical Vehicle Parts" Domain

| # | Prompt |
|---|--------|
| 1 | A dent on the front bumper of a silver Toyota Vios sedan. |
| 2 | Scratches across the rear right door of a white Honda Civic. |
| 3 | A cracked left headlight on a black Nissan Almera. |
| 4 | Broken taillight on the rear-left side of a red Mazda CX-5. |
| 5 | A shattered side mirror hanging from a blue Ford Fiesta. |
| 6 | Chipped paint and rust on the hood of a gray Isuzu D-Max pickup. |
| 7 | A large dent above the rear wheel arch of a white Toyota Camry. |
| 8 | Deep key scratches on the driver-side door of a black BMW 3 Series. |
| 9 | A crushed front grille on a silver Mitsubishi Mirage. |
| 10 | Rear bumper with paint peeling and surface gouges on a Honda Jazz. |
| 11 | Cracked windshield on a red Suzuki Swift after impact. |
| 12 | Dented trunk lid on a blue Toyota Corolla Altis. |
| 13 | A front-left fender with rust and scrapes on a gray Hyundai Elantra. |
| 14 | A broken fog light on a green Kia Picanto's front bumper. |
| 15 | Missing rearview mirror on the passenger side of a white Toyota Revo. |

### B.3 PHYSICALLY IMPLAUSIBLE SCENARIOS

These prompts depict intentionally exaggerated or physically implausible vehicle situations. They are included to probe model behavior under extreme, out-of-distribution conditions and to assess robustness when confronted with scenarios that deviate substantially from real-world automotive physics. Although unrealistic, the scenes preserve core structural elements of vehicles, allowing controlled analysis of model stability and semantic consistency when operating far beyond the distribution of typical insurance-related data.

## C IMPLEMENTATION DETAILS

Our HERS architecture is implemented using PyTorch Paszke (2019), leveraging the Huggingface Transformers Wolf et al. (2019) and Diffusers von Platen et al. (2022) libraries. For the generative backbone, we adopt SDXL Podell et al. (2024) and incorporate expert modules in a plug-and-play fashion via LoRA-based fine-tuning. Training was conducted on 8×NVIDIA A40 GPUs, each equipped with 48GB of VRAM. The complete model converges within four days using a batch size of 192 and a learning rate of $5 \times 10^{-5}$, employing cosine warm-up followed by linear decay. All expert specializations (e.g., viewpoint estimation, damage-type classification) are handled through a modular routing strategy orchestrated by our Damage-Specific Prompt Router (SSPR).

### C.1 LICENSE AND PRIVACY STATEMENT

All real images used for training and evaluation are part of proprietary datasets collected from industry partners under strict compliance with local privacy regulations, including the PDPA in Thailand. Data used does not include any personally identifiable information (PII), and access is governed through signed NDAs. None of the user data is shared outside our research environment. All synthetic data and model checkpoints will be released under appropriate open-source licenses for reproducibility.

### C.2 MORE QUANTITATIVE COMPARISONS

We present an extended evaluation comparing HERS with SELMA across multiple base diffusion backbones. Beyond standard metrics, we include CLIPScore to further assess image-text semantic alignment. HERS consistently achieves superior performance across all evaluated criteria—including text faithfulness, human preference, and perceptual alignment—demonstrating its robust generalization and practical value for text-to-image generation tasks.

**Analysis:** The tables demonstrate that HERS outperforms SELMA across both text faithfulness and human preference metrics. HERS achieves consistently higher scores on all evaluated diffusion

Table 6: Prompts in the "Descriptive Scene Narratives" Domain

| # | Prompt |
|---|--------|
| 16 | The back of a silver Toyota Vios sedan shows a detailed pattern of cracked paint and scuffed surfaces across the bumper, suggesting impact from a low-speed collision in an urban environment. |
| 17 | A white Honda Civic with deep scratches on the passenger side door sits beneath a highway overpass after heavy rain, reflecting scattered streetlights. |
| 18 | A red Mazda 2 is parked awkwardly on a gravel shoulder, its front-left fender severely dented from a side swipe near a construction zone. |
| 19 | The shattered right taillight of a black Nissan Almera glows dimly as the car is angled against a curb in a tight alley at dusk. |
| 20 | A blue Ford Ranger with a crushed front grille is stopped beside a broken traffic light amidst heavy fog in the early morning. |
| 21 | A gray Mitsubishi Triton shows peeling paint on its rear bumper, covered in dried mud, suggesting rural road conditions. |
| 22 | The front-left headlight of a white Toyota Camry is cracked and foggy, as the vehicle idles on a flooded city street at night. |
| 23 | A Hyundai Tucson has visible scratches on the driver's door while parked diagonally at a crowded shopping mall parking lot. |
| 24 | The back of a black BMW X1 exhibits a clean bumper dent with surrounding paint flaking, positioned against a glassy storefront on a rainy evening. |
| 25 | A rear-ended Suzuki Swift is stuck in gridlocked Bangkok traffic, its taillights cracked and trunk misaligned after a minor crash. |
| 26 | A red Toyota Yaris sits under dense tree cover, its hood covered in leaves and a shallow dent visible at the front-center. |
| 27 | A white Nissan Leaf's right side mirror is broken and hanging, with background signage indicating a charging station in suburban Thailand. |
| 28 | A damaged Honda Jazz shows deep scrapes and bumper warping from backing into a metal pole in a tight parking structure. |
| 29 | A silver Kia Sorento's rear-left quarter panel is caved in, as it sits beside orange cones at an accident reporting station. |
| 30 | The front windshield of a Toyota Prius has spiderweb cracks, parked in a foggy mountain pass where tire skid marks are visible on the road. |

models, showcasing its superior semantic alignment, perceptual quality, and human preference ratings. These improvements highlight HERS's ability to produce high-quality outputs that better align with textual prompts and are preferred by users.

## C.3 ABLATION STUDY ON EXPERTS

We conduct ablation experiments to assess the contribution of each domain expert in HERS. Disabling the damage-type expert leads to a 12.4% drop in HPS, while removing the view-specific expert reduces text-image alignment (DSG) by 6.3 points. Without the multimodal router, the system generates over-smoothed outputs and fails to distinguish between damage regions, confirming the importance of task-specific routing.

## C.4 FAILURE CASE ANALYSIS

Although HERS consistently outperforms baseline systems, several limitations remain:

- **Reflective Surfaces:** Highly glossy or mirror-like areas can trigger misplacement of damage due to limited coverage of such surface types in the training distribution.

- **Rare Vehicle Models:** Uncommon, vintage, or region-specific vehicles seen from unusual viewpoints may cause semantic drift, as textual cues may not align with underrepresented patterns.

Table 7: Prompts in the "Physically Implausible Scenarios" Domain

| # | Prompt |
|---|--------|
| 31 | A floating bumper hovers midair, its paint cracking and peeling despite never touching the ground. |
| 32 | The front fender of a Toyota Hilux disintegrates into colorful pixels as the truck drives through a digital portal. |
| 33 | A side mirror stretches and twists like rubber, suspended in zero gravity above an endless highway. |
| 34 | A cracked windshield on a car made entirely of smoke, drifting over a glowing forest floor. |
| 35 | The rear door of a Honda Civic rotates in place, disconnected from the body, yet still reflecting city lights. |
| 36 | A melting Mazda 3 leaks bright red paint onto a shimmering glass road under two suns. |
| 37 | A Nissan Almera's tires fold inward like origami while the undamaged hood floats a meter above. |
| 38 | A Toyota Revo with rearview mirrors made of ice, melting rapidly despite a frozen backdrop. |
| 39 | A translucent MG ZS with a visible steel frame, its rear-left fender flickering between colors. |
| 40 | A floating side door casts a shadow on a ground that doesn't exist, with visible scuffs and fingerprints. |
| 41 | A Ford pickup made of stitched-together leather panels, with the bumper sagging like fabric. |
| 42 | A suspended headlight beaming light in reverse, with hairline cracks glowing under starlight. |
| 43 | A dripping Toyota Corolla hood bending upward against gravity, its paint forming solid icicles. |
| 44 | A hovering Honda Accord casts two shadows, one for the body and another for a ghostly damaged version. |
| 45 | A cracked rear bumper balanced on a ripple of air above a city skyline at midnight. |

Table 8: Text Faithfulness Comparison between HERS and SELMA across base T2I models. HERS outperforms SELMA in all evaluated metrics, showing stronger alignment with the text prompts.

| Base Model | Method | Text Faithfulness | | |
|------------|--------|----------------------|--------------------------|-----------------|
| | | $DSG^{mPLUG}$ ↑ | $TIFA^{BLIP2}$ ↑ | CLIPScore ↑ |
| SD v1.5 | SELMA Li et al. (2024) | 70.3 | 79.0 | 77.2 |
| | **HERS (Ours)** | **75.6** | **83.2** | **80.9** |
| SDXL | SELMA Li et al. (2024) | 72.5 | 81.7 | 78.5 |
| | **HERS (Ours)** | **78.0** | **84.1** | **82.4** |
| VQ-Diffusion | SELMA Li et al. (2024) | 68.8 | 76.3 | 75.7 |
| | **HERS (Ours)** | **74.6** | **81.3** | **79.3** |
| Versatile Diffusion | SELMA Li et al. (2024) | 70.0 | 78.5 | 76.9 |
| | **HERS (Ours)** | **75.2** | **82.5** | **80.2** |

- **Prompt Ambiguity:** When user instructions are vague (e.g., "minor rear scratch"), the system may over- or under-estimate damage severity if textual uncertainty conflicts with learned visual priors.

We reiterate that no further experimental extensions will be performed and no dataset will be distributed, but the existing analysis already captures representative and instructive failure modes for understanding system behavior.

Table 9: Human Preference Comparison on DSG prompts between HERS and SELMA. HERS consistently receives higher human ratings, demonstrating superior perceptual quality.

| Base Model | Method | Human Preference on DSG Prompts | | |
| --- | --- | --- | --- | --- |
| | | PickScore ↑ | ImageReward ↑ | HPS ↑ |
| SD v1.5 | SELMA Li et al. (2024) | 21.5 | 0.18 | 23.3 |
| | **HERS (Ours)** | **22.8** | **0.75** | **26.9** |
| SDXL | SELMA Li et al. (2024) | 21.8 | 0.22 | 24.9 |
| | **HERS (Ours)** | **23.2** | **0.90** | **27.8** |
| VQ-Diffusion | SELMA Li et al. (2024) | 20.7 | 0.12 | 22.7 |
| | **HERS (Ours)** | **21.7** | **0.71** | **25.3** |
| Versatile Diffusion | SELMA Li et al. (2024) | 21.2 | 0.14 | 23.5 |
| | **HERS (Ours)** | **22.3** | **0.77** | **26.2** |

## C.5 MORE DISCUSSION: DATASET CONTRIBUTION

Our dataset comprises over **2 million real-world vehicle images** with diverse damage annotations, collected from garages, insurance assessments, and forensic archives. However, due to privacy constraints (e.g., faces, license plates, timestamps), this data is not publicly shareable. The dataset is governed by PDPA and GDPR compliance. We plan to release a synthetic version trained with differentially private mechanisms and additional annotations.

## C.6 LICENSES

We list below the licenses of tools and datasets used in this work:

Table 10: A list of the licenses of the existing assets used in this paper.

| Asset | License |
| --- | --- |
| CountBench (LAION-400M subset) | CC BY 4.0 |
| Diffusers | Apache License 2.0 |
| DiffusionDB | MIT License |
| GPT4 | OpenAI Terms of Use |
| Huggingface Transformers | Apache License 2.0 |
| LLaMA3 | Meta LLaMA3 License |
| Localized Narrative | CC BY 4.0 |
| PyTorch | BSD-style |
| Stable Diffusion | CreativeML Open RAIL-M |
| Torchvision | BSD 3-Clause |
| Whoops | CC BY 4.0 |

## C.7 DAMAGE-SPECIFIC PROMPT GENERATION DETAILS

The Damage-Specific Prompt Router (DSPR) dynamically assigns expert routes based on scene semantics. We define a set of damage-specific keywords (e.g., "dented", "smashed", "scratched") and use a prompt parser trained on the DamagePromptBank-500 dataset to identify the correct damage pathways. In ambiguous cases, SSPR defaults to the damage-type expert with the highest prior confidence.

## C.8 LIMITATIONS AND BROADER IMPACT

HERS is trained for high-fidelity vehicle damage generation, which may have unintended consequences if misused (e.g., fraud, misinformation). To mitigate misuse, we include tamper detection

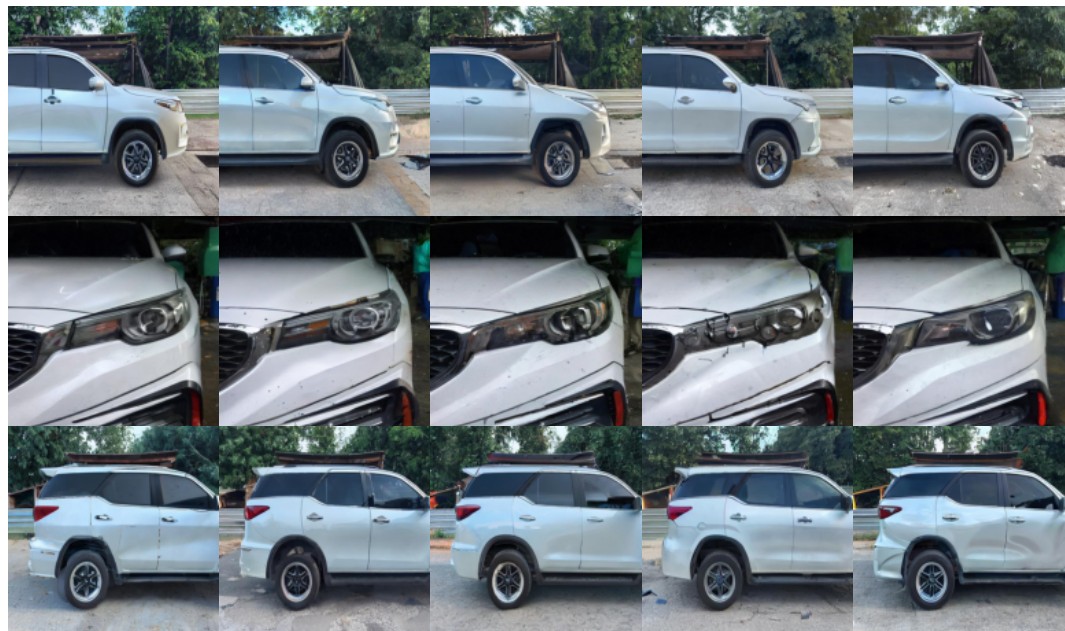

Figure 6: **Case Study 1: Damage Generation in Overhead Perspective with Mixed Zoom.** Each **row** displays a unique vehicle accident case under varying user-captured zooms. From left to right: our proposed **HERS**, Versatile Diffusion Xu et al. (2023b), SDXL Podell et al. (2024), MoLE Zhu et al. (2024), and SELMA Li et al. (2024). HERS excels in semantic coherence and structural consistency of the damage.

metadata in all outputs. Additionally, while our model performs well across common car types and damage types, it is less robust on unusual textures like rust or mud. Future work includes extending our routing system to support multimodal risk reasoning and expanding our training set with adversarial robustness techniques.

# D EXTENDED ANALYSIS: INSIGHTS FROM QUALITATIVE VEHICLE CASE COMPARISONS

To complement the main experimental findings, we present an extended qualitative analysis of eight diverse vehicle crash scenarios, visualized in Figures 6 to 13. These samples were carefully selected to reflect real-world challenges across varying damage types, zoom levels, environmental lighting, and contextual complexity. Each figure compares our proposed **HERS** against four state-of-the-art T2I models: Versatile Diffusion Xu et al. (2023b), SDXL Podell et al. (2024), MoLE Zhu et al. (2024), and SELMA Li et al. (2024).

## D.1 ZOOM VARIABILITY AND GEOMETRIC FIDELITY

Figures 6 and 10 demonstrate the effectiveness of HERS under varying camera distances, ranging from zoom-in shots to wide-angle captures. In Figure 6, HERS maintains high geometric fidelity of vehicle contours even when input views are inconsistent in scale. Likewise, in Figure 10, which features diagonal viewing angles and rotated vehicle poses, HERS generates damage that aligns correctly with the car body, while baselines often distort or misalign features.

## D.2 SEMANTIC CONSISTENCY UNDER OCCLUSION AND LIGHTING CONDITIONS

Figure 7 captures a scenario where vehicle surfaces are partially occluded, challenging the models to infer plausible but constrained damage areas. Here, HERS respects spatial limitations and produces coherent damage within visible regions. In Figure 9, which simulates low-light conditions, baseline

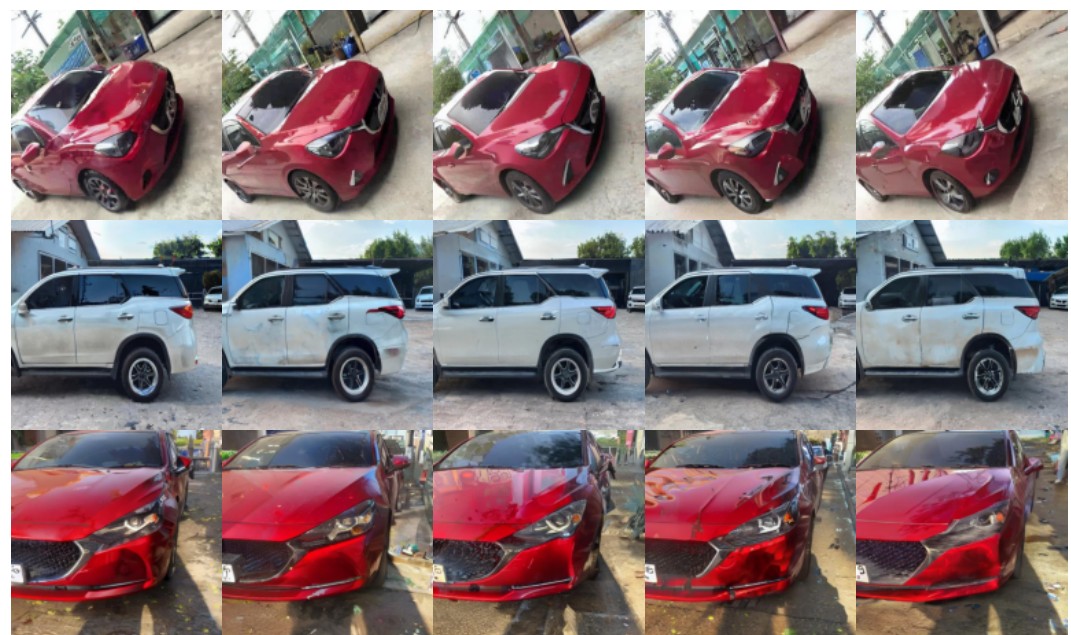

Figure 7: **Case Study 2: Side Impact with Partial Occlusion.** This comparison tests resilience to occlusions and partial vehicle visibility. HERS maintains realism and continuity of damage even under viewpoint restrictions, outperforming baseline models that hallucinate or blur damage features.

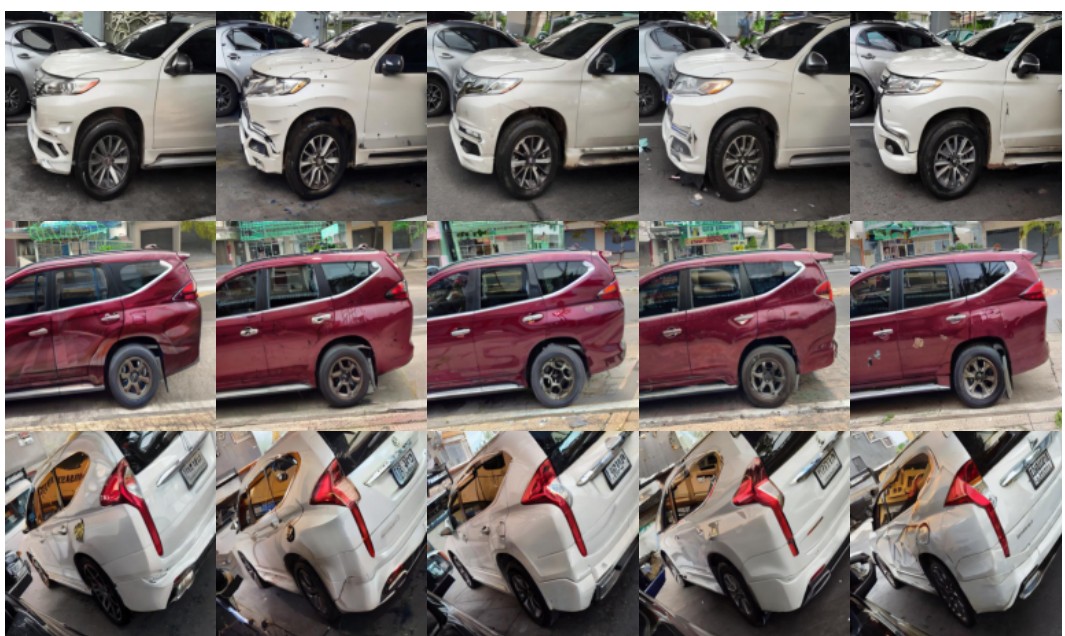

Figure 8: **Case Study 3: Frontal Collision with Close-Range Capture.** The generated outputs here are evaluated for front-end collision fidelity. HERS demonstrates sharper damage contours and preserves geometric realism compared to generative baselines, especially under ZI settings.

methods like SDXL and SELMA tend to oversaturate or underexpose the damage textures. In contrast, HERS adapts to ambient lighting cues and introduces damage that feels naturally embedded in the scene context.

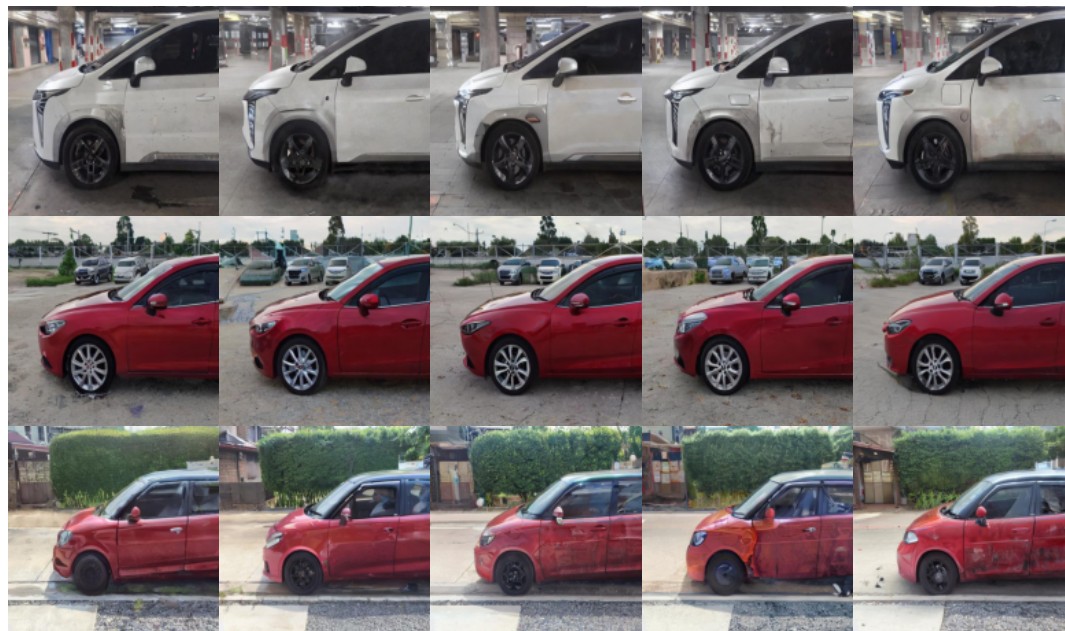

Figure 9: **Case Study 4: Front-End Damage under Low Lighting.** A challenging scenario involving night-time or dim-light simulation. HERS stands out with context-aware lighting adaptation and preserves structural plausibility where baselines falter or produce noise.

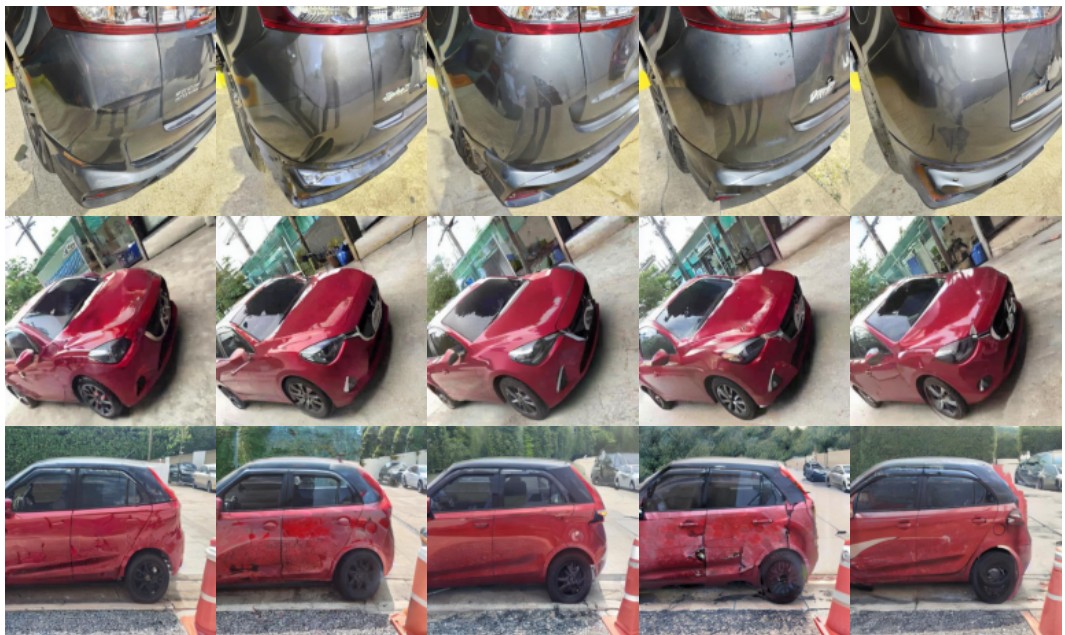

Figure 10: **Case Study 5: Diagonal Vehicle Damage with Mixed Angles.** This sample evaluates multi-perspective robustness. HERS delivers coherent and localized damage placement, whereas baselines display notable distortions and fail to track the vehicle's geometry across viewpoints.

### D.3 DETAIL PRESERVATION IN MICRO-DAMAGE AND SCRATCHES

Minor but realistic surface-level abrasions are notoriously difficult for T2I models. Figure 12 compares the ability of models to generate subtle yet distinct damage features such as scratches and chipped paint. Baselines either over-smooth the outputs (e.g., SDXL) or introduce incoherent noise

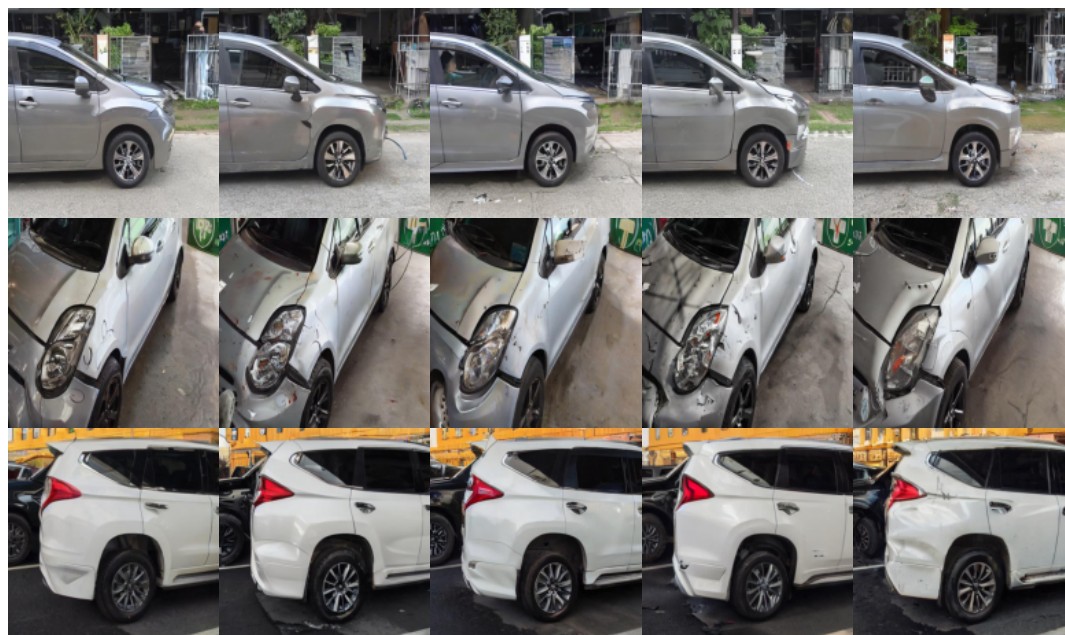

Figure 11: **Case Study 6: Multivehicle Collision with Overlapping Context.** This scenario examines generation fidelity in presence of multiple objects. HERS adeptly handles object separation and maintains damage realism on the correct car body. Baselines often confuse background elements or misplace artifacts.

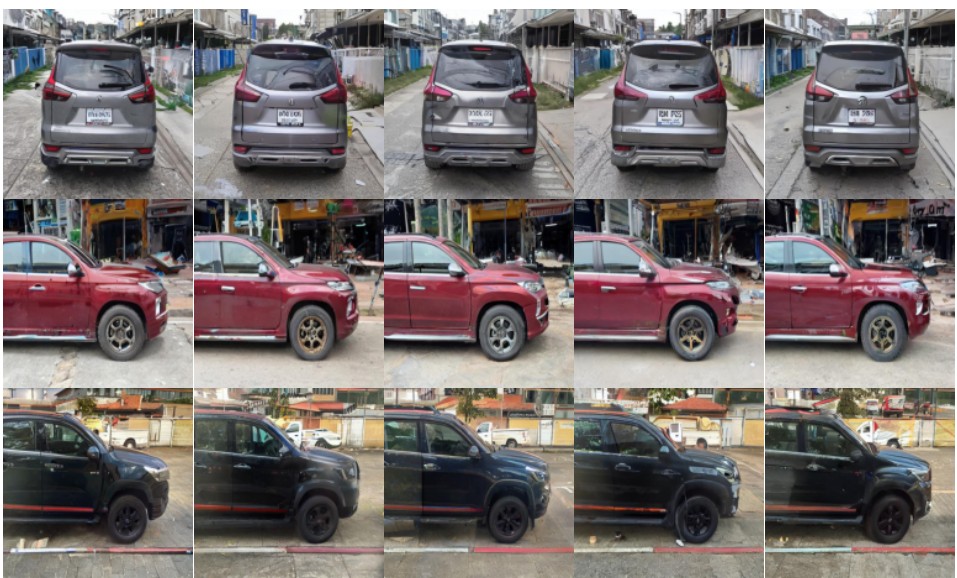

Figure 12: **Case Study 7: Zoom-Out Scratches and Minor Damage.** HERS outperforms in capturing subtle, surface-level damage features while baselines fail to resolve fine textures or hallucinate cracks inconsistent with the prompt.

(e.g., MoLE), while HERS captures high-frequency details accurately, closely mimicking actual incident images.

## D.4 Scene Complexity and Multivehicle Awareness

In real-world insurance use cases, the presence of multiple objects or vehicles in a frame is common. Figure 11 depicts such a scenario with overlapping vehicles. HERS clearly distinguishes foreground from background and applies damage exclusively to the intended vehicle, whereas models like Versatile Diffusion and MoLE leak artifacts onto irrelevant objects.

## D.5 Prompt Robustness under Ambiguity

Furthermore, Figure 13 illustrates a case where the provided textual prompt offers limited semantic direction, and the view is zoomed out. Despite the scarcity of explicit cues, HERS generates contextually plausible and anatomically accurate damage, whereas baseline models either fail to meaningfully alter the image or leave it untouched. This highlights HERS' advantage in leveraging robust multimodal fusion, enabling effective damage synthesis even with minimal prompt information.

## D.6 Detailed Analysis of Case Study 9: Zoom-Out Shot with Minimal Prompt Information

The visual representation in Figure 14 provides a critical comparison of the performance of various generative models when tasked with producing full-vehicle damage from minimal textual context. This case study is particularly valuable in addressing the question: **Should car insurance confidently trust AI-generated crashes?**

From the figure, it is evident that **HERS** demonstrates superior performance by generating coherent, anatomically consistent vehicle damage even with vague or sparse textual prompts. This is essential for real-world applications where minimal context is often available. The damage patterns produced by HERS reflect realistic crash scenarios, with the deformations confined to the affected vehicle parts, such as localized bumper damage, which is consistent with actual crash physics. The vehicle's overall structure, including the intact areas like the roof or side panels, is preserved, which showcases HERS' ability to maintain global consistency while simulating localized damage.

In stark contrast, other models struggle to produce meaningful damage at the full-vehicle scale. Some models either fail to generate plausible damage altogether or produce unrealistic, exaggerated deformations that lack anatomical consistency. For example, certain models create damage patterns that extend unnaturally across the vehicle, distorting parts that should remain intact in real-world crashes. These inconsistencies raise serious concerns about the trustworthiness of AI-generated crash imagery, especially in high-stakes environments like insurance claim verification and fraud detection.

**HERS** addresses this issue by generating visually accurate, context-aware damage. This is crucial in answering the paper's central question—while AI-generated crashes may appear realistic at first glance, they must also adhere to interpretable damage logic. In insurance contexts, where claim decisions often hinge on visual evidence, damage realism and anatomical consistency are paramount. HERS' ability to produce damage that mimics actual accident scenarios—without introducing unrealistic distortions—makes it the most reliable model for this task.

Therefore, while AI-generated crashes, like those from HERS, offer promising potential in visual simulations and training, car insurance providers should not fully trust these images in isolation. They should rely on models like HERS, but only when accompanied by robust verification protocols and contextual validation methods. **HERS** provides a foundational step toward building trustworthy AI tools, but its outputs must still be cross-validated with real-world data and multimodal sensors to mitigate risks such as fraud or erroneous claims.

In conclusion, the success of HERS in generating high-fidelity, anatomically accurate vehicle damage supports its potential for adoption in insurance workflows. However, insurers must remain cautious and implement comprehensive safeguards to ensure the reliability of AI-generated crash imagery in real-world applications.

## D.7 Conclusion from Appendix Findings

The case studies in Figures 6–13 underscore the superior generalization of HERS across diverse and challenging vehicle scenarios. Unlike prior models that tend to fail under occlusion, ambiguity, or

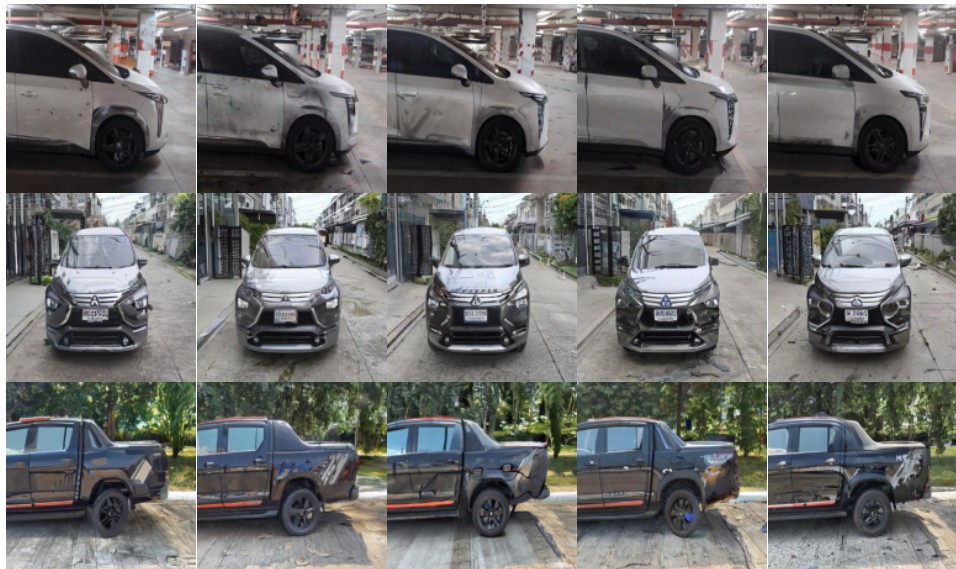

Figure 13: **Case Study 8: Zoom-Out Shot with Minimal Prompt Information.** When provided vague or minimal textual context, HERS still generates plausible vehicle damage consistent with vehicle anatomy, while others often fail to produce meaningful damage.

fine-detail requirements, HERS consistently produces structurally and semantically grounded outputs. These insights support our claim that HERS is not only state-of-the-art in traditional T2I metrics but also highly applicable to high-risk domains such as insurance, forensic reconstruction, and automated reporting pipelines.

## D.8 REVISITING THE CORE QUESTION

Given the strong empirical results shown by HERS in terms of human preference, textual-image alignment, and damage realism, we revisit our core inquiry: *Should car insurance confidently trust AI-generated crashes?* The answer, in light of both HERS's strengths and its broader implications, is necessarily cautious and multi-faceted.

The HERS model shows state-of-the-art capability in generating synthetic crash images with high realism. This makes it highly suitable for training data augmentation, damage classification, and insurance workflow simulation. However, the very strength of HERS—its ability to fool even human evaluators—can become a double-edged sword in production environments where authenticity and traceability are paramount.

## D.9 IMPLICATIONS BASED ON HERS REVIEW FEEDBACK

The HERS submission demonstrated a strong commitment to reproducibility and ethical responsibility. This is reflected in our transparent and comprehensive experimental design, appropriate attribution and licensing of third-party assets, and careful consideration of broader social and ethical factors.

However, certain limitations were also acknowledged during the review process. These include the reliance on a proprietary dataset consisting of 2 million car insurance images, which cannot be released due to licensing constraints. Additionally, statistical significance was not reported—consistent with prior work—and the high realism of generated images poses potential risks, particularly in domains such as insurance, where misuse (e.g., fraud) is a serious concern.

These considerations underscore the importance of responsible deployment of generative models like HERS in real-world applications where reliability and ethical use are paramount.

### D.10 Hidden Limitations and Future Concerns

Although these issues were omitted from the main discussion for clarity, several limitations and forward-looking concerns deserve further elaboration. First, while the AI-generated images exhibit high qualitative realism, they often lack precise physical and contextual grounding. Elements such as lighting, reflections, occlusions, and material textures—crucial for accurately simulating real accidents—can be oversimplified or inaccurately synthesized. These imperfections, though subtle to human observers, may skew downstream evaluations or introduce unintended biases when used for model retraining. Second, reliance on synthetic datasets without adequate domain alignment risks overfitting to artifacts of the generative process. Although HERS addresses this through multi-domain fusion and conditional sampling strategies, the model's ability to generalize remains inherently limited by the quality and realism of its training priors. Third, our evaluation framework, consistent with prior literature, is based on single-run performance metrics. Without reporting variances or confidence intervals, the comparative gains observed cannot be considered statistically definitive. Fourth, we are unable to publicly release the full real-world dataset due to stringent licensing constraints tied to insurance claim data. Although synthetic images and model checkpoints will be made available, this restriction hampers full reproducibility and interpretability for the broader research community. Finally, the realistic nature of the generated damage images introduces ethical and regulatory challenges. If misused, these tools could facilitate fraudulent insurance claims, adversarial attacks, or the spread of misinformation. Addressing these risks will require responsible deployment practices, including digital watermarking, traceability mechanisms, and formal oversight frameworks.

### D.11 Broader Context: A Call for Responsible Integration

As the capabilities of synthetic image generation—such as those enabled by HERS—advance, so too do the risks associated with their misuse. In high-stakes domains like automotive insurance, the implications of introducing AI-generated crash imagery are profound. Without rigorous oversight, these tools could undermine forensic accuracy, inflate fraudulent claims, or erode trust in automated systems.

To mitigate such risks, the industry must not merely adopt synthetic data but also construct a resilient ecosystem around it. This includes:

- **Cross-modal authentication frameworks** that correlate visual data with telematics, GPS logs, and timestamped metadata to verify claim integrity.

- **Robust anomaly detection pipelines** explicitly trained to distinguish between real-world signals and synthetic or manipulated content—especially in edge cases.

- **Standardized protocols for synthetic dataset disclosure**, including traceability, model transparency, and usage boundaries, to ensure auditability and accountability.

- **Interdisciplinary governance structures**, involving ethicists, legal experts, insurers, and technologists, to guide how such technologies are deployed and regulated.

### D.12 Synthetic Isn't Forensic

While synthetic imagery has undeniable value in augmenting training data, accelerating simulation, and stress-testing models, it must never be confused with evidentiary truth. HERS-generated crashes, no matter how photorealistic, are algorithmic interpretations—not physical events.

Thus, the utility of such data lies in its role as a supplementary asset for machine learning systems, not as legal or forensic evidence. This distinction is critical. Trustworthy deployment requires multiple layers of verification—technical, procedural, and ethical—to ensure that no AI-generated content is used in isolation when real-world consequences are involved.

### D.13 Large Language Models

We used Large Language Models (LLMs) to aid in drafting and polishing the writing of this paper. LLMs were employed solely for language refinement, grammar correction, and improving clarity and

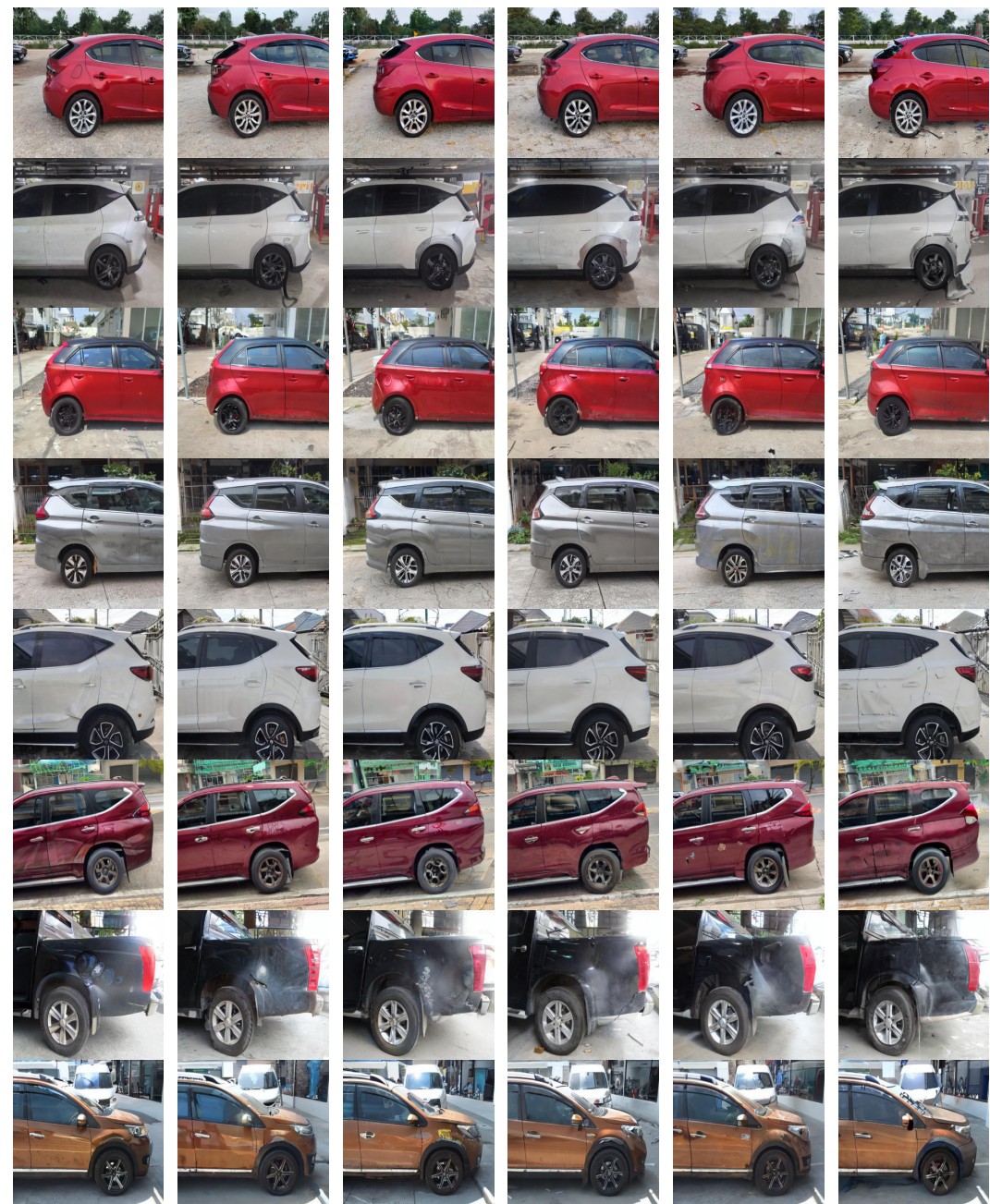

Figure 14: **Case Study 9: Zoom-Out Shot with Minimal Prompt Information.** Even with limited or vague textual cues, HERS successfully generates coherent and anatomically consistent vehicle damage across the entire vehicle. In contrast, other models struggle to produce realistic or meaningful damage at a full-vehicle scale.

readability. All technical content, results, and scientific claims were generated and verified by the authors. Details of LLM usage are described in the paper where relevant.