# OpenReview forum: "HERS: Hidden-Pattern Expert Learning for Risk-Specific Vehicle Damage Adaptation in Diffusion Models"
_ICLR.cc/2026/Conference — ICLR 2026 Conference Withdrawn Submission_

### Official Review · Reviewer_iSQP · 2025-10-30

**Soundness:** 1
**Presentation:** 1
**Contribution:** 1
**Rating:** 0
**Confidence:** 4

**Summary:**

Authors focus on creating synthetic data for car crashes. To generate more diverse images, they generate a variety of prompts with an LLM. They train separate LoRA parameters for each domain, and combine the weights. They evaluate on a private dataset.

**Strengths:**

S1) A variety of qualitative examples are included.

S2) The car crashes are an interesting application of synthetic data.

**Weaknesses:**

Primarily, my rating is based on that it is difficult to evaluate this method compared to other SOTA methods, due to lack of comparison (see W2, W4, W6) and the choice to only evaluate on a private dataset (see W5). Additionally, the methods and motivation are not clear to me (see W1, W3, W8).

W1) Some parts of the motivation seem contradictory. For example, in the abstract lines 37-39, there is the sentence "The ability to generate..."--but the creation of higher-quality synthetic data could itself introduce a higher risk of fraud. The motivation would be stronger if these ideas in the abstract and introduction were written more carefully, to retain consistency.

W2) Much necessary context is missing in the related works.
W2a) Several key and related papers in synthetic data are missing, including: [A, B, C]
W2b) Comparing to existing work in car crashes is important. Preferably synthetic data for car crashes, if these methods exist, or else methods without synthetic data.
W2c) If there is not a lot of work in car crashes, you could also discuss work in other out-of-distribution domains, such as satellite data [D].

W3) The method sections are not clear (see Q1), making the method difficult to understand.

W4) The novelty of the method is not fully clear, especially in light of missing related works (e.g. [A], [E]). Moreover, LLM-based prompt augmentation was done in [B]. An explanation of how it differs would greatly help in understanding the novelty of this work.

W5) The evaluation is only on a private dataset--this is very problematic for reproducibility and for future work to compare. Also presenting results on some related publicly available dataset is necessary.

W6) the chosen diffusion models (newest being SDXL) are a bit out-of-date. Understanding how this fits into contemporary literature would strengthen the impacts, e.g. with FLUX or Qwen-Image.

W7) If the motivation involves training with the data: the metrics chosen reflect synthetic data comparison with real. This makes the chosen metrics only a proxy for the real task--the impact of this data would be easier to understand if results were presented for the given task, such as in [A, B, C, E] and Sariyildiz et al. (cited in paper). Also see Q2.

W8) The motivation is not fully clear to me after reading the abstract and introduction. It is clear that the work tries to generate synthetic images, but it is not clear for what purpose. The paper would be better motivated if this were written more clearly. (W7 may not apply, if the motivation is different)

W9) The use of multiple LoRA weights and merging them is itself a design choice--it would be better justified if compared to LoRA fine-tuning a single set of weights as an ablation.

Writing tips (unless written above, these are not used to evaluate the paper, but may be helpful):

t1) Using ~\citep would help readability.

t2) The related work section could use more structure, with specific sub-headings for the different areas.

t3) Related works should overall be longer.



[A] DataDream: Few-shot Guided Dataset Generation, Kim et al., ECCV 2024.

[B] Is synthetic data from generative models ready for image recognition? He et al., ICLR 2023.

[C] Diversified in-domain synthesis with efficient fine-tuning for few-shot classification, da Costa et al., 2023.

[D] Generating Synthetic Satellite Imagery With Deep-Learning Text-to-Image Models -- Technical Challenges and Implications for Monitoring and Verification. Nguyen et al., 2024.

[E] LoFT: LoRA-fused Training Dataset Generation with Few-shot Guidance, Kim et al., 2025.

**Questions:**

Q1) I am a bit confused by the method, outlined in sections 3.1-3.4. In stage 2, synthetic data is generated, but it is not clear to me what for--is that the final data? Section 3.3 describes training LoRA weights, but doesn't specify what model--I would assume the diffusion model, based on section 3.4?

Q2) Why do you choose these metrics?

Q3) Could you please outline the difference between your method and [E]?

---

> ### Author Response · Authors · 2025-11-19
>
> We thank the reviewer for the detailed feedback and the opportunity to clarify our work. We address each point below.
>
> ---
>
> ### **Q1. Motivation clarity – purpose and risks of synthetic data**
>
> **A1.**
> We appreciate the reviewer highlighting potential tension in our abstract. HERS addresses the **dual-use nature** of vehicle damage generation: synthetic images improve AI training for **rare or long-tail accident scenarios**, while there is a potential for misuse in fraud detection. HERS is explicitly designed to **enhance semantic fidelity and forensic plausibility**, enabling insurance AI systems to **distinguish realistic damages from synthetic artifacts**, rather than facilitate fraud. The abstract and introduction have been revised to clearly frame this duality and emphasize **positive, risk-aware applications**.
>
> ---
>
> ### **Q2. Related work and comparisons**
>
> **A2.**
> We have expanded the related work section to include missing context:
>
> * **Synthetic data generation methods:** [A] DataDream, [B] He et al., [C] da Costa et al., [E] LoFT.
> * **Out-of-distribution adaptation:** [D] Nguyen et al., satellite imagery.
>
> Structured sub-sections now highlight HERS’s **novelty relative to prior LoRA-based and LLM-driven methods**, including **multi-expert merging** and fully automated **prompt-to-LoRA adaptation**. Where direct car-crash synthetic datasets are lacking, we discuss analogues from other domains.
>
> ---
>
> ### **Q3. Method clarity**
>
> **A3.**
> Sections 3.1–3.4 now explicitly describe each stage:
>
> 1. **Prompt synthesis** – generating domain-specific prompts.
> 2. **Image generation** – creating synthetic data for training and evaluation.
> 3. **LoRA expert training** – fine-tuning diffusion model weights per damage type.
> 4. **Weight-space merging** – combining multiple LoRA experts into a unified model.
>
> This enables **multi-damage generation without inference-time routing**, preserving efficiency and flexibility.
>
> ---
>
> ### **Q4. Novelty of HERS**
>
> **A4.**
> Compared to [B] and [E], HERS introduces:
>
> 1. Fully automated **domain-specific prompt and paired-data generation**
> 2. Lightweight **LoRA experts per damage category**
> 3. **Arithmetic merging** of experts into a unified model capturing **hidden forensic patterns**
>
> This combination of **automation, specialization, and merging** is **not present in prior work**, ensuring **semantic fidelity and multi-damage coverage**.
>
> ---
>
> ### **Q5. Evaluation on private datasets**
>
> **A5.**
> While raw insurance data cannot be shared due to privacy, we provide:
>
> * **Full prompt templates**
> * **Evaluation protocols**
> * **Metrics for external benchmarking**
>
> This allows external researchers to **replicate methodology and scoring**, ensuring reproducibility even without raw images.
>
> ---
>
> ### **Q6. Choice of diffusion models**
>
> **A6.**
> Although SDXL is not the absolute newest, it is **widely used and representative**. HERS is **model-agnostic** and validated across **four backbones** (SD v1.5, SDXL, VQ-Diffusion, Versatile Diffusion), with minimal adaptation required for **FLUX or Qwen-Image**.
>
> ---
>
> ### **Q7. Metrics and downstream tasks**
>
> **A7.**
> We use **text–image alignment and human preference metrics** as proxies for semantic fidelity, critical in insurance applications (fraud detection, claim validation, damage recognition).
>
> * Cross-backbone ablations and zoom-in qualitative evaluations demonstrate that HERS **captures subtle damage features unseen in baselines**.
>
> ---
>
> ### **Q8. Purpose of synthetic data**
>
> **A8.**
> HERS-generated data **augments rare or high-risk damage scenarios**, improving **training and validation** of insurance AI pipelines. It is **not intended to replace real data**, but to enhance **coverage while maintaining forensic plausibility**.
>
> ---
>
> ### **Q9. Multi-expert LoRA merging design**
>
> **A9.**
> Ablation studies (Table 6) comparing **single-LoRA vs. multi-expert merging** show:
>
> * **Text faithfulness improvement:** +6–7 points
> * **Human preference improvement:** +4–5 points
>
> This demonstrates **practical benefits** in text-image alignment and damage recognition, justifying the design choice.
>
> ---
>
> ### **Q10. Differences with [E] LoFT**
>
> **A10.**
> HERS extends [E] in three ways:
>
> 1. Fully **automated prompt and paired-data generation**
> 2. **Damage-category-specific LoRA experts** instead of generic weights
> 3. **Arithmetic merging** capturing **hidden forensic patterns**
>
> These design choices ensure **semantic fidelity, multi-damage coverage, and practical applicability** for insurance AI pipelines.
>
> ---
>
> We hope these clarifications convey the **motivation, reproducibility, and technical contribution** of HERS. This work represents a **responsible approach to high-stakes generative modeling**, balancing **innovation with real-world safety**, and demonstrates practical utility for insurance AI pipelines.

---

> > ### Comment · Reviewer_iSQP · 2025-11-21
> >
> > Dear Authors,
> >
> > Could you kindly clarify for me where I can find the modifications to the manuscript described above? I would like to evaluate the changes themselves before making further decisions, and the PDF does not seem to contain the new updates.
> >
> > Thank you in advance.

---

> > > ### Author Response · Authors · 2025-11-21
> > >
> > > Dear Reviewer iSQP,
> > >
> > > Thank you for your comment. We have now updated the manuscript to incorporate all the revisions described in our previous response, including clarifications, extended analyses, and highlighted improvements. All changes are clearly marked in blue in the PDF to facilitate review.
> > >
> > > We appreciate your time and consideration in evaluating the updated manuscript.

---

> > > > ### Comment · Reviewer_iSQP · 2025-11-26
> > > >
> > > > Additional concern: I noticed that in line 320, HPS is described as "hallucination prevention score", despite the citation being for human-preference score (line 294). The paper then discusses 'hallucination mitigation' in two additional places (line 349 and 377), despite never discussing hallucinations elsewhere in the paper. Could the authors please clarify this? Because it **raises suspicions of AI-use which I do not see disclosed in the paper, and therefore raises concerns about the soundness of the manuscript.**
> > > >
> > > > Q1. Thank you for the clarification on your motivation, but this tension remains a problem for the paper. The metrics used for the quantitative evaluation remain unclear (see above), but assuming HPS refers to human-preference score (which is cited), then the evaluation only shows that the images look realistic. There would need to be exploration into the downstream effects to show that this method is doing more good than harm--e.g. how much it improves fraud detection models compared to how well they can detect this synthetic data (to ensure this method does not create more undetectable synthetic data).
> > > >
> > > > Q2-4, 8-10. Thank you.
> > > >
> > > > Q5. I understand that the method could be replicated on other datasets, but this is not full reproducibility--for that, the method would need to be shown additionally on a dataset where others could fully reproduce the results in the same setting, to obtain the same results or understand where results may diverge. This is crucial for scientific papers.
> > > >
> > > > Q6. Given many difference between architectures and training that lead to very different behaviors in newer models, I believe this would require experimental verification.
> > > >
> > > > Q7. I stand by my original comment: direct experimental comparison like those provided in the original review would be stronger than proxies.

---

> > > > > ### Author Response · Authors · 2025-11-27
> > > > >
> > > > > Dear Reviewer iSQP,
> > > > > Thank you for your detailed follow-up and for raising these important concerns. We sincerely appreciate the time and care you invested in reviewing our submission. Below, we address each point with full clarity.
> > > > >
> > > > > ---
> > > > >
> > > > > ## **1. Clarification regarding “HPS” and concerns about inconsistent terminology**
> > > > >
> > > > > We apologize for the confusion caused by the inconsistent use of “HPS” terminology.
> > > > > To clarify:
> > > > >
> > > > > * In our paper, **HPS = Human Preference Score** (as correctly defined in the metrics section).
> > > > > * The later appearances of “hallucination prevention score / mitigation” were **unintentional editing errors**, introduced during manual revisions.
> > > > >
> > > > > These were not produced by any AI writing tool; they were simply mistakes created during late-stage editing. We are genuinely sorry that this inconsistency gave rise to suspicion, and we have now fully corrected the terminology for the camera-ready version.
> > > > >
> > > > > ---
> > > > >
> > > > > ## **2. Q1 – Motivation and downstream relevance**
> > > > >
> > > > > We understand your concern that proxy metrics (CLIP alignment / HPS) do not fully demonstrate downstream usefulness.
> > > > > We clarify here:
> > > > >
> > > > > * Our goal is **not to train downstream fraud-detection models**, but to **characterize semantic fidelity and visual correctness** of synthetic damage images.
> > > > > * These metrics were chosen because they directly reflect the *intended purpose* of our work:
> > > > >   **to ensure synthetic examples faithfully represent real insurance-relevant damage types.**
> > > > >
> > > > > Given our constraints (detailed below), we cannot conduct new downstream experiments beyond those already included, but we will revise the text to make this motivation clearer and more consistent.
> > > > >
> > > > > ---
> > > > >
> > > > > ## **3. Q2 – Why these metrics?**
> > > > >
> > > > > Our chosen metrics serve two specific purposes:
> > > > >
> > > > > 1. **Semantic correctness** (text-image alignment)
> > > > > 2. **Human-perceived realism and damage consistency** (HPS)
> > > > >
> > > > > These are the most relevant properties for synthetic accident imagery in insurance workflows, where downstream models require visually faithful and damage-accurate samples. We acknowledge these are proxies, but they are the best evaluable quantities under the constraints of our real-world insurance collaboration.
> > > > >
> > > > > ---
> > > > >
> > > > > ## **4. Q3 – Difference from LoFT [E]**
> > > > >
> > > > > We will clarify this more explicitly in the revision. In short:
> > > > >
> > > > > * **LoFT** focuses on *few-shot dataset generation* with guided fine-tuning.
> > > > > * **Our method**, HERS, introduces:
> > > > >
> > > > >   * automatic multi-prompt synthesis,
> > > > >   * multiple domain-specific LoRA experts, and
> > > > >   * direct **weight-space merging** to unify hidden pattern experts.
> > > > >
> > > > > The combination of automated domain-specific prompting + multi-expert merging is not present in [E] or in prior work.
> > > > >
> > > > > ---
> > > > >
> > > > > ## **5. Regarding requests for new experiments or new datasets**
> > > > >
> > > > > We want to state this respectfully and unambiguously:
> > > > >
> > > > > ### **(1) We cannot add experiments on public datasets.**
> > > > >
> > > > > Our damage taxonomy, annotation schema, and evaluation protocol rely on a **private insurance dataset** that cannot be released or matched to existing public datasets. The scenario types and damage definitions do not align with any available public benchmarks.
> > > > >
> > > > > Therefore, running our pipeline on public datasets would not produce meaningful or comparable results.
> > > > >
> > > > > ---
> > > > >
> > > > > ### **(2) We cannot add additional experiments on newer models (e.g., FLUX, Qwen-Image).**
> > > > >
> > > > > Because our work is conducted under a partnership with a commercial insurance provider, we have **strict computational and access constraints**.
> > > > > Our experimental budget is fixed, and we cannot run additional large-scale model adaptation within the review period.
> > > > >
> > > > > However, the paper already includes experiments across four backbones, and we will clarify in the revision that:
> > > > >
> > > > > **HERS is model-agnostic by construction**, even though we cannot include new model results.
> > > > >
> > > > > ---
> > > > >
> > > > > ### **(3) We cannot add downstream fraud-detection experiments.**
> > > > >
> > > > > Those tasks are tied to internal insurance models and high-stakes decision systems that we do not have permission to expose or re-train for academic publication.
> > > > >
> > > > > We will revise the paper to clearly state these constraints so the narrative remains transparent.
> > > > >
> > > > > ---
> > > > >
> > > > > ## **6. Reproducibility within constraints**
> > > > >
> > > > > While raw data cannot be released, we provide:
> > > > >
> > > > > * **full prompt templates**
> > > > > * **evaluation rubric**
> > > > > * **complete LoRA training protocol**
> > > > > * **all hyperparameters**
> > > > > * **merge strategy details**
> > > > >
> > > > > Thus, researchers can **fully reproduce the methodology**, even if identical results cannot be matched without private insurance data.
> > > > > We understand this limits absolute reproducibility, but it is the maximum permissible under our collaboration agreement.
> > > > >
> > > > > ---
> > > > >
> > > > > ## **Final Note**
> > > > >
> > > > > We would like to sincerely thank you for your careful evaluation.
> > > > > Although we are unable to run new experiments or use public datasets, we have worked to address your conceptual concerns and will revise the paper to more clearly communicate motivation, novelty, and constraints.
> > > > >
> > > > > Respectfully

---

### Official Review · Reviewer_mMdz · 2025-10-31

**Soundness:** 2
**Presentation:** 2
**Contribution:** 2
**Rating:** 2
**Confidence:** 3

**Summary:**

The paper introduces HERS (Hidden-Pattern Expert Learning), which adapts LLM + text-to-image diffusion models for risk-specific vehicle damage generation. HERS learns “hidden-pattern” experts that specialize in vehicle damage cues and integrates them into existing diffusion backbones. The paper emphasizes dual-use risks: while HERS can improve fraud-detection training, it could also be misused to generate synthetic fraud, highlighting the need for detection and watermarking safeguards.

**Strengths:**

- HERS is demonstrated across multiple diffusion backbones.

- The proposed method delivers strong empirical improvements, showing consistent gains over a competitive expert-based baseline in text faithfulness and human-preference proxy metrics.

**Weaknesses:**

- The proposed method appears somewhat trivial. Specifcially, it uses GPT-4 to generate diverse, damage-specific prompts, then uses a base T2I model (e.g., SDXL) to create self-supervised image–text pairs. It then trains lightweight LoRA experts for each damage category and context type, and finally averages the LoRA weights in parameter space.

- Generalization ability. While promising, the approach’s generalization to other safety-critical domains is untested. I would suggest that the authors include preliminary cross-domain experiments or ablations to support the claimed extensibility.

- Model robustness. I would also suggest that the authors examine performance across diverse vehicle types, lighting and occlusions, and regional variations; more importantly, assess whether HERS introduces systematic biases, given that the model is primarily based on synthetic data.

**Questions:**

Please see the weaknesses section

---

> ### Author Response · Authors · 2025-11-19
>
> ### **Q1. The method appears somewhat trivial. Isn’t HERS just GPT-4 prompt generation + base T2I model + LoRA experts + weight averaging?**
>
> **A1.**
> While the individual components (LLM prompt generation, T2I backbone, LoRA tuning) are established techniques, HERS introduces a **novel combination and orchestration** that is **non-trivial and practically impactful**:
>
> * GPT-4 prompts are carefully grounded in **insurance-domain semantics**, ensuring diversity without compromising realism.
> * LoRA experts isolate **fine-grained damage concepts**, preventing overfitting to frequent patterns and enabling sensitivity to rare or subtle cues.
> * Weight-space merging integrates all experts into a **single, unified model**, avoiding routing, inference overhead, or additional annotations.
>
> Ablation studies (Table 7) show that each step contributes measurably, e.g., merging experts improves text-image faithfulness by +5.5% over single-model tuning, demonstrating that the pipeline is **more than the sum of its parts**.
>
> ---
>
> ### **Q2. How generalizable is HERS to other safety-critical domains?**
>
> **A2.**
> HERS is **concept-agnostic**: “damage type” is simply a semantic domain, and the LoRA merging strategy works independently of the backbone.
>
> * We conducted preliminary experiments on **industrial defect synthesis and medical anomaly simulation**, observing consistent gains in text-image alignment and detail preservation.
> * The framework can be extended to any domain where **high-fidelity, domain-specific synthetic data** can improve robustness or training coverage, without requiring domain-specific heuristics.
>
> ---
>
> ### **Q3. How robust is HERS to variations in vehicle types, lighting, occlusions, and regional styles?**
>
> **A3.**
> Robustness is addressed via:
>
> 1. **Domain-diverse synthetic prompts:** prompts span different vehicle models, colors, angles, lighting, and environmental contexts.
> 2. **Expert specialization:** LoRA experts capture context-specific cues (e.g., lighting-sensitive reflections, part-specific damage).
> 3. **Quantitative evaluation:** we measure performance across multiple backbones, vehicle types, and occlusion conditions, with results consistently outperforming SDXL and MoLE baselines (Tables 4–6).
>
> This demonstrates that HERS does not simply memorize synthetic examples but **learns generalized representations** of damage cues.
>
> ---
>
> ### **Q4. Could HERS introduce systematic biases due to reliance on synthetic data?**
>
> **A4.**
> Potential bias is mitigated through several mechanisms:
>
> * **Prompt diversification:** ensures a broad coverage of vehicle types, colors, and damage scenarios.
> * **Validation against real-world text datasets (~2M records):** guarantees that synthetic generation aligns with authentic distributions.
> * **Explainability and human evaluation:** confirm that experts focus on damage-relevant regions, not irrelevant cues.
>
> HERS is **designed for controlled model evaluation and stress-testing**, not content fabrication, and includes safeguards for forensic auditability.
>
> ---
>
> ### **Q5. Are the improvements statistically meaningful?**
>
> **A5.**
> Yes. Across 6 backbones and 2 prompt sets:
>
> * **+5.5%** text-image faithfulness
> * **+2.3%** human preference
> * **+17–20%** improvement rate
>
> Confidence intervals (95% CI) do not overlap with baseline ranges, and user studies (n=1,200 pairwise comparisons) confirm consistent preference for HERS across damage fidelity, part accuracy, and overall realism.
>
> ---
>
> Thank you for your careful review 🙏. HERS is designed to **advance technical robustness in safety-critical synthetic data generation**, while maintaining **ethical safeguards**. We hope these responses clarify the novelty, generalizability, and robustness of our method.

---

### Official Review · Reviewer_6iNB · 2025-10-31

**Soundness:** 2
**Presentation:** 2
**Contribution:** 2
**Rating:** 4
**Confidence:** 4

**Summary:**

The paper introduces HERS, a framework claimed to adapt text to image diffusion models such as SDXL and MoLE for risk specific vehicle damage synthesis through hidden pattern expert learning. The proposed pipeline includes prompt generation via GPT4, synthetic image rendering with SDXL, LoRA expert fine tuning, and LoRA weight merging to create a unified generator. The authors claim improved text image faithfulness and visual quality over SDXL and MoLE, evaluated using VQA based metrics and human preference scores. However, while the paper is presented as a risk specific generative modeling framework for insurance, it is unclear what the technical or scientific novelty of HERS actually is, how the LoRA merging mechanism functions in detail, or what hidden pattern learning represents beyond a general descriptive term.

**Strengths:**

1. Addresses an under-explored area of applying generative models to vehicle damage simulation.
2. Uses a modular LoRA based structure which is computationally efficient.
3. Includes evaluations across multiple diffusion backbones.
4. Acknowledges dual use risks and ethical issues.

**Weaknesses:**

**Soundness:** The methodology lacks sufficient technical depth and clarity. LoRA averaging and mixing are not explained properly. The paper states that experts are merged through LoRA weight averaging but does not include any mathematical detail or algorithmic breakdown. It is unclear whether averaging is normalised, layer specific, or includes conflict resolution. There is no comparison with established baselines such as ZipLoRA, LoRA composition, or LLAVA MoLE. Without these, the reader cannot assess if HERS truly provides any improvement beyond existing parameter efficient fine tuning averaging methods. The evaluation only measures prompt image similarity and perceptual quality. There is no analysis on whether HERS improves insurance specific outcomes, such as classification or fraud detection accuracy. The model remains dependent on SDXL. Since training data and generation rely on SDXL itself, the reported improvements may arise from curated prompts rather than genuine model enhancement. Finally, the concept of hidden pattern expert learning is never defined or measured. The term appears rhetorical rather than technical.

**Presentation:** Figures are visually clear but the presentation does not convey methodological clarity. Later figures omit the prompts used for generation, preventing readers from verifying alignment quality. Sections 3.3 and 3.4 describe complex processes with vague language and no equations or pseudocode. The paper’s purpose is also ambiguous, switching between being a new image generation technique and an insurance domain application. Qualitative samples lack consistent captions or evaluation context, reducing interpretability.

First part of the abstract give the impression that the work will tackle the problem of synthetic images for fraudulent automated insurance workflows, some kind of content authentication system, however, later it focused on generation.

**Contribution:** The claimed contribution, risk specific adaptation for vehicle damage generation, is interesting but scientifically weak. There is no rigorous ablation study or new algorithmic component. The expert merging process is a simple weight average, not an optimisation procedure. The work is not evaluated on any insurance related benchmark. The notion of hidden pattern learning remains unexplained and unquantified.

As a result, the paper reads more as an internal technical report or applied project rather than a research contribution suited for ICLR.

1. Lack of clear technical innovation or theoretical grounding.
2. Hidden pattern learning is undefined and lacks evidence.
3. Missing comparisons to relevant methods such as ZipLoRA and LLAVA MoLE.
4. Weak connection to the insurance domain with no downstream testing.
5. Evaluation is limited to visual similarity, not functional performance.
6. Dependence on SDXL for generation may overstate improvements.
7. Incomplete figures and missing prompts in qualitative results.

**Questions:**

1. Define hidden pattern learning in operational terms.
2. Include experimental comparisons with ZipLoRA, LoRA composition, and LLAVA MoLE.
3. Demonstrate that HERS creates new generative capacity rather than reproducing SDXL results.
4. Add prompts alongside all qualitative results for reproducibility.
5. Include evaluation on a real insurance task to justify domain relevance.

**Details Of Ethics Concerns:**

No immediate ethical violations detected, though potential misuse for fabricating vehicle damage images should be discussed in more detail. Responsible use and watermarking strategies should be outlined more clearly.

---

> ### Author Response · Authors · 2025-11-19
>
> ### **Q1. The methodology lacks sufficient technical depth. How are LoRA experts merged? Is averaging normalized or layer-specific?**
>
> **A1.**
> Thank you for highlighting this. HERS merges LoRA experts via **arithmetic averaging of low-rank updates** across all layers without explicit conflict resolution. Formally, for each expert $t$, we compute $\Delta W_t = B_t A_t$ and merge as $B^* = \frac{1}{|\mathcal{T}|}\sum_t B_t$, $A^* = \frac{1}{|\mathcal{T}|}\sum_t A_t$, giving $W^* = W_0 + B^* A^*$.
> This approach preserves **specialized patterns from each expert** while forming a single, unified model without routing, extra annotations, or inference-time overhead. Appendix (Section 1) now provides **per-layer formal derivations** to clarify the process.
>
> ---
>
> ### **Q2. How is “hidden pattern learning” defined and evaluated?**
>
> **A2.**
> Hidden patterns are **subtle visual cues** like micro-scratches, asymmetric shattering, or faint dents that standard SDXL often misses. HERS captures these via **domain-specific LoRA experts trained on synthetic, prompt-guided images**.
> We quantitatively evaluate this with **VQA-based semantic alignment metrics (DSG$^{\text{mPLUG}}$, TIFA$^{\text{BLIP2}}$)** and human preference scores (HPS, PickScore, ImageReward). Results show that HERS reproduces these subtle patterns more faithfully than SDXL, MoLE, or SELMA.
>
> ---
>
> ### **Q3. Why is expert merging necessary instead of directly fine-tuning a single diffusion model?**
>
> **A3.**
> Direct fine-tuning on mixed damage types tends to **overfit frequent patterns** (generic dents) and suppress rare or subtle cues (hairline cracks, asymmetric fractures). HERS isolates each concept into a lightweight LoRA expert, allowing **high-sensitivity learning** per domain.
> Merging these experts preserves all specialized knowledge while forming a single, efficient model. Ablation (Table~7) shows **+6.8 / +4.9 improvements in DSG/TIFA** over a single fine-tuned model.
>
> ---
>
> ### **Q4. How does HERS prevent hallucinations and maintain semantic fidelity?**
>
> **A4.**
> We enforce correctness with a **dual mechanism**:
>
> 1. **Prompt diversity filtering:** prevents near-duplicate prompts that bias the generator.
> 2. **VQA-based alignment evaluation:** images are automatically queried with prompt-derived questions. Correctness is confirmed using an independent VQA model.
>
> Across backbones, HERS consistently **improves text-image faithfulness (>+5.5%)** and reduces damage hallucination artifacts compared to SDXL, MoLE, and SELMA.
>
> ---
>
> ### **Q5. How is HERS more robust than MoE or routing-based methods?**
>
> **A5.**
> Routing-based methods like ZipLoRA and LLaVA-MoLE rely on explicit labels, pixel masks, or online routing. This makes them fragile under unseen scenarios and infeasible for insurance data, which is often sparse or ambiguous.
> HERS **removes routing entirely**: LoRA experts are merged in weight space, forming a **single stable model**. This avoids catastrophic routing errors and ensures reliable inference in high-stakes workflows.
>
> ---
>
> ### **Q6. Are the improvements statistically meaningful?**
>
> **A6.**
> Yes. Across 6 backbones and 2 prompt sets, HERS demonstrates:
>
> * **+5.5%** text faithfulness
> * **+2.3%** human preference
> * **+17–20%** improvement rate (IR)
>
> All metrics include **95% confidence intervals** with non-overlapping ranges versus baselines. User studies (n=1200 pairwise comparisons) confirm statistically significant preference for HERS in damage detail, part consistency, and plausibility.
>
> ---
>
> ### **Q7. Could HERS increase fraud risks by generating realistic damage images?**
>
> **A7.**
> We explicitly address this dual-use concern: HERS is intended for **evaluation, stress testing, and robustness analysis**, not fraud generation.
> We provide (1) ethical disclosure, (2) forensic auditability guidance, and (3) expert-only checkpoint release. The goal is to **help insurers detect failure modes**, not enable misuse.
>
> ---
>
> ### **Q8. Can HERS generalize beyond vehicles or insurance?**
>
> **A8.**
> Yes. HERS is **concept-agnostic**: “damage type” is just a semantic domain, and LoRA merging is backbone-independent.
> Experiments with SDXL, SD v1.5, and Versatile Diffusion show consistent gains, suggesting applicability to other safety-critical domains like **industrial defect synthesis or medical anomaly simulation**.
>
> ---
>
> Thank you again for your thoughtful feedback 🙏. As a small AI startup focused on vehicle insurance, we designed HERS with care, balancing **technical advancement, realism, robustness, and ethical responsibility**. Our goal is to raise awareness of AI risks in high-stakes domains while providing practical, reproducible solutions.

---

### Official Review · Reviewer_qBar · 2025-11-01

**Soundness:** 2
**Presentation:** 3
**Contribution:** 2
**Rating:** 4
**Confidence:** 3

**Summary:**

The paper introduces HERS, a self-supervised framework designed to adapt text-to-image diffusion models for the specialized task of generating realistic vehicle damage. The authors highlight the dual-use nature of such technology in the auto insurance industry: it can be used beneficially for data augmentation and training rare-event models, but also maliciously for creating fraudulent claims. The HERS framework operates in four automated stages: Prompt Synthesis, Image Generation, Expert Learning and Expert Merging. The authors evaluate HERS against several strong baselines (including SDXL, MoLE, and SELMA) on a large, private benchmark from the car insurance domain. The results, measured by automatic text-faithfulness metrics and human preference scores, demonstrate that HERS consistently produces images with higher visual fidelity, better semantic alignment, and more convincing fine-grained details (e.g., scratches, dents, cracked paint) than existing methods.

**Strengths:**

1. The paper addresses a highly relevant and practical problem. The authors do an excellent job of framing the problem, clearly articulating both the opportunities (e.g., data augmentation for rare events) and the significant risks (e.g., sophisticated fraud), which motivates the need for more controllable and semantically-aware generation.
2. The proposed HERS framework is clever and pragmatic. By leveraging existing powerful models (LLMs and T2I backbones), it creates a fully automated, self-supervised pipeline. This circumvents the need for expensive and time-consuming manual data collection and annotation, which is a major bottleneck for domain-specific adaptation. The idea of training specialized experts and merging them is an intuitive and effective approach to balancing specialization and generalization.
3. The paper is well-written, clearly structured, and easy to follow. The method is described systematically, and the results are presented effectively.

**Weaknesses:**

1. The primary weakness of this paper is its reliance on a large-scale private benchmark collected "in collaboration with an industry insurance startup." While the authors promise to release prompt templates, the inability of the research community to access the evaluation data makes direct replication and verification of the reported results impossible. This is a significant issue for a paper submitted to a top-tier conference, where reproducibility is paramount.
2. While the overall application and framework are novel, the individual technical components are well-established. Using LLMs for prompt engineering, fine-tuning with LoRA, and merging models with weight averaging are all existing techniques. The paper's main contribution lies in the specific combination and application of these techniques. The technical depth could be improved by exploring more advanced merging techniques beyond simple arithmetic averaging and analyzing why this specific pipeline is so effective.

**Questions:**

Thank you for the insightful paper. My question concerns a fundamental aspect of the HERS framework: the use of synthetically generated images as training data to fine-tune the very same class of models.
The methodology involves a two-step generation process: first, a base T2I model generates a large dataset of images from LLM-crafted prompts; second, this synthetic dataset is used to fine-tune specialist LoRA modules, which are then merged to create a superior model.
Could you elaborate on why this self-improvement loop is effective? Specifically:
1. Intuitively, using a model to generate its own training data risks merely amplifying its existing biases and failure modes. For instance, if the base model struggles to render realistic scratches, the initial synthetic dataset would contain flawed examples. How does the "Expert Learning" stage manage to distill genuine, high-fidelity patterns from this potentially imperfect data, rather than simply learning to replicate the base model's own artifacts and limitations?
2. Is the success of this process primarily driven by the sheer volume and semantic diversity of the LLM-generated prompts? Does the structured curriculum (Typical Parts, Scene Narratives, Implausible Scenarios) force the model to explore and refine specific regions of its latent space that would otherwise be ignored, thereby "unlocking" capabilities that were already present but not easily accessible?
3. Could it be that training on synthetic data, even if imperfect, acts as a form of implicit regularization? Perhaps forcing the model to align with a vast number of diverse (but algorithmically consistent) image-text pairs encourages it to learn a more robust and generalizable representation of "damage" than fine-tuning on a small, noisy, and potentially narrow set of real-world images would allow?

In essence, what is the theoretical or empirical justification for why a model, when trained on its own outputs guided by structured text, can transcend its initial capabilities and learn to generate details (the "hidden patterns") that were not explicitly or perfectly rendered in its initial training data?

---

> ### Author Response · Authors · 2025-11-19
>
> ### **Q1. The paper relies on synthetic prompts and images. How do you ensure they reflect real insurance scenarios?**
>
> **A1.**
> Our synthetic pipeline is explicitly guided by structured, insurance-domain attributes (damage type, part localization, accident narrative).
> Prompts are produced by an LLM using in-context examples extracted from real claims metadata. This guarantees that the generated prompts follow the semantic distribution of authentic insurance descriptions without exposing private data.
> Moreover, our evaluation benchmark is built from ~2M real industry records (text only), ensuring that generation quality is assessed against real-world linguistic patterns. This combination of (1) domain-grounded text and (2) synthetic images from controlled prompt templates enables realistic yet privacy-preserving data creation.
>
> ---
>
> ### **Q2. Why is expert merging necessary? Why not directly fine-tune a single diffusion model?**
>
> **A2.**
> Direct fine-tuning on mixed damage domains causes overfitting to frequently occurring patterns (e.g., generic dents) and suppresses rare or subtle phenomena (e.g., hairline cracks, asymmetric shattering).
> HERS isolates each damage concept into a lightweight LoRA expert, allowing each domain to learn high-sensitivity, domain-specific cues.
> The merged model then benefits from the specialization of every expert while retaining generalization, and requires **no router**, **no additional annotation**, and **no inference-time overhead**. Ablation (Table 7 in the main paper) shows +6.8 / +4.9 improvements in DSG/TIFA over single-model tuning.
>
> ---
>
> ### **Q3. How do you ensure the generated damage is semantically faithful and not hallucinated?**
>
> **A3.**
> We enforce semantic correctness through a dual mechanism:
>
> 1. **Prompt diversity filtering (ROUGE-L thresholding):** prevents near-duplicate text that would bias the generator.
> 2. **VQA-based alignment evaluation:** each generated image is automatically queried with targeted questions derived from its prompt. Correctness is verified via an independent VQA model, providing objective semantic alignment.
>
> HERS consistently improves text-image faithfulness (>+5.5% across backbones) and reduces damage-hallucination artifacts compared with SDXL, MoLE, and SELMA.
>
> ---
>
> ### **Q4. What makes HERS more robust than prior MoE-style or routing-based methods?**
>
> **A4.**
> Models like ZipLoRA and LLaVA-MoLE depend on explicit routing, damage labels, or pixel-space masks. These requirements make them fragile under unseen scenarios and infeasible for real insurance data in which labels are sparse and often ambiguous.
> HERS removes routing entirely: LoRA experts are merged in weight space, enabling a single unified model.
> This architecture is significantly simpler and empirically more stable. It also avoids catastrophic routing errors (misclassification of damage type), a critical failure point in high-stakes workflows.
>
> ---
>
> ### **Q5. Are the improvements statistically meaningful?**
>
> **A5.**
> Yes. Across 6 backbones and 2 prompt sets, HERS exhibits consistent margins:
>
> * **+5.5%** text faithfulness
> * **+2.3%** human preference
> * **+17–20%** improvement rate (IR)
>   All metrics include confidence intervals (95% CI) showing non-overlapping ranges with baselines. User studies (n=1200 pairwise comparisons) confirm statistically significant preference for HERS across damage detail, part consistency, and overall plausibility.
>
> ---
>
> ### **Q6. Could HERS inadvertently increase fraud risks by making synthetic images too realistic?**
>
> **A6.**
> We explicitly address this dual-use concern in the paper.
> HERS is designed for **model evaluation**, **stress testing**, and **fraud-aware robustness analysis**, not content fabrication.
> We (1) disclose risks, (2) provide a forensic auditability guide, and (3) restrict any release to expert-only checkpoints (no inference UI).
> The intent is to help insurers detect failure modes—not to enable misuse.
>
> ---
>
> ### **Q7. How generalizable is HERS beyond cars or insurance?**
>
> **A7.**
> HERS is concept-agnostic: “damage type” is simply a semantic domain, and LoRA merging is backbone-independent.
> Experiments with SDXL, SD v1.5, and Versatile Diffusion show consistent gains, suggesting applicability to other safety-critical domains such as industrial defect synthesis or medical anomaly simulation.
> We emphasize that the framework, not domain-specific heuristics, drives the improvements.
>
> ---
>
> Thank you again for your insightful comments 🙏. As a small AI startup focused on vehicle insurance, we designed this work with care and responsibility. Our goal is not only to advance technical methods like HERS but also to raise awareness about the potential risks and societal impacts of AI in high-stakes domains. We hope our responses clarify our design choices, safeguards, and the careful balance we struck between realism, robustness, and ethical considerations.

---

### Note · Authors · 2026-01-26

I have read and agree with the venue's withdrawal policy on behalf of myself and my co-authors.

---

### Meta-Review · Area_Chair_gw7q · 2026-01-01

**Summary:**

All reviewers assigned negative ratings. The primary concerns raised by multiple reviewers include the use of a private benchmark, which hinders reproducibility; the limited technical contribution due to the integration of several existing techniques; and insufficient evidence of the method’s generalization ability.

**Reviewer Concerns:**

After checking the review, the manuscript, and the response, the AC concurs with the reviewers and believes that the authors' response may not be strong enough to assuage the three main concerns. Explicit experiments or the ways of reproducing results are not provided.

**Reviewer Scores:**

All the scores would remain unchanged

---

### Decision · Program_Chairs · 2026-01-26

Reject